# The ER tether VAPA is required for proper cell motility and anchors ER-PM contact sites to focal adhesions

**Hugo Siegfried[†], Georges Farkouh, Rémi Le Borgne, Catherine Pioche-Durieu, Thaïs De Azevedo Laplace[‡], Agathe Verraes, Lucien Daunas, Jean-Marc Verbavatz\*, Mélina L Heuzé\***

Université Paris Cité, CNRS, Institut Jacques Monod, F-75013, Paris, France

**\*For correspondence:**
jean-marc.verbavatz@ijm.fr (J-MarcV);
melina.heuze@ijm.fr (MLH)

**Present address:** [†]Cell Polarity, Migration and Cancer Unit, Université Paris Cité, UMR3691 CNRS, Institut Pasteur, Paris, France; [‡]Université Paris Cité, INSERM, Centre de Recherche sur l'Inflammation, F-75018, Paris, France

**Competing interest:** The authors declare that no competing interests exist.

**Abstract** Cell motility processes highly depend on the membrane distribution of Phosphoinositides, giving rise to cytoskeleton reshaping and membrane trafficking events. Membrane contact sites serve as platforms for direct lipid exchange and calcium fluxes between two organelles. Here, we show that VAPA, an ER transmembrane contact site tether, plays a crucial role during cell motility. CaCo2 adenocarcinoma epithelial cells depleted for VAPA exhibit several collective and individual motility defects, disorganized actin cytoskeleton and altered protrusive activity. During migration, VAPA is required for the maintenance of PI(4)P and PI(4,5)P2 levels at the plasma membrane, but not for PI(4)P homeostasis in the Golgi and endosomal compartments. Importantly, we show that VAPA regulates the dynamics of focal adhesions (FA) through its MSP domain, is essential to stabilize and anchor ventral ER-PM contact sites to FA, and mediates microtubule-dependent FA disassembly. To conclude, our results reveal unknown functions for VAPA-mediated membrane contact sites during cell motility and provide a dynamic picture of ER-PM contact sites connection with FA mediated by VAPA.

## Editor's evaluation

This manuscript presents important findings that bring together two important topics in cell biology: the function of membrane contact sites and cell migration. The authors present compelling evidence to support a role of the ER tether protein VAP-A in focal adhesion dynamics and cell motility. This paper will be of interest to those cell biologists and biophysicists working on adhesion, migration, and membrane contact site biology.

## Introduction

The lipid identity of membrane compartments is highly regulated within cells and plays a key role in a diversity of cellular processes. During cell motility, lipids, in particular Phosphoinositides (PInst), are local determinants of molecular events leading to cell polarization, the formation of actin-driven protrusions and turnover of focal adhesions (FA) (*Hammond and Burke, 2020*; *Tsujita and Itoh, 2015*). PI(4,5)P2, the most abundant PInst at the plasma membrane, controls FA dynamics by regulating the binding of talin to integrins (*Martel et al., 2001*; *Thapa et al., 2012*) and the polarized trafficking of integrins (*Nader et al., 2016*). PI(4,5)P2 is described as a modulator of actin cytoskeleton organization and dynamics, either directly through the recruitment of several actin-binding proteins (*Senju et al., 2017*), or indirectly by regulating the activation of RhoA (*Lacalle et al., 2007*; *Xu et al., 2010*) and Cdc42 GTPases (*Daste et al., 2017*). The product of PI(4,5)P2 phosphorylation, PI(3,4,5)P3, also plays a crucial role in cell motility at the plasma membrane (PM). Several in vitro and in vivo

studies have shown that PI(3,4,5)P3 acts as a 'compass lipid' that stabilizes the direction of migration (*Funamoto et al., 2002*; *Hannigan et al., 2002*; *Lam et al., 2012*) through the activation of Rac1 (*Kunisaki et al., 2006*; *Welch et al., 2002*; *Yoshii et al., 1999*).

Membrane lipids are transported through vesicular transport, but also through non-vesicular transport at so-named membrane contact sites (MCS; *Jackson et al., 2016*). MCS are sites of close apposition between two membranes, often the endoplasmic reticulum (ER) membrane and the membrane of another organelle, at a distance of less than 80 nm, where exchanges of lipids, Ca2 +and metabolites take place (*Prinz et al., 2020*; *Scorrano et al., 2019*; *Wu et al., 2018*). While the activity of lipid transfer at MCS was observed decades ago (*Vance, 1990*), their implication in patho-physiological processes such as cell motility has gained interest only recently (*Prinz et al., 2020*).

In this work, we studied the role of VAPA, a tethering protein at ER-mediated MCS during cell motility. VAPA is a member of the highly conserved VAP family of proteins, together with VAPB. VAP proteins are integral ER membrane proteins bridging the ER membrane to the membrane of other compartments by assembling with numerous partners including lipid transfer proteins that bind the target membrane, such as Nir proteins and OSBP-related (ORP) proteins. The assembly of most of VAP complexes relies on the interaction between the conserved N-terminal MSP (Major Sperm Protein) domain of VAPs and the FFAT (2 phenyl alanines in an acidic tract) motif on their partner. VAPs also contain a central coiled-coil dimerization domain and an ER transmembrane domain in their C-terminus (*Murphy and Levine, 2016*). VAP proteins are involved in several cellular functions such as lipid transport, membrane trafficking, the unfolded protein response pathway and microtubules organization (*Kamemura and Chihara, 2019*; *Lev et al., 2008*). VAP complexes with lipid transfer proteins control the homeostasis of PInst and sterols at various intracellular locations. At ER-PM contact sites, the VAP/Nir2 and VAP/ORP3 complexes regulate the transport of Phosphoinositol (PI) and PI(4) P respectively. Together with membrane-associated kinases and phosphatases, they contribute to modulating locally the pool of PI(4)P, PI(4,5)P2 and PI(3,4,5)P3 at the PM (*Chang and Liou, 2015*; *Gulyás et al., 2020*; *Kim et al., 2013*). In addition, the association of VAPs with OSBP-related proteins modifies the level of PI(4)P at endosomal membranes and at the Golgi, therefore regulating trafficking events in these two compartments (*Dong et al., 2016*; *Kawasaki et al., 2022*; *Mesmin et al., 2013*). Moreover, VAP proteins mediate cholesterol transport at ER-Golgi, ER-endosome and ER-peroxisome MCS (*Dong et al., 2019*; *Hua et al., 2017*; *Mesmin et al., 2013*; *Wilhelm et al., 2017*). Depending on the cell type and organism, the loss of VAP proteins can lead to either a reduction (*Peretti et al., 2008*) or an accumulation (*Mao et al., 2019*; *Wakana et al., 2021*; *Subra et al., 2023*) of PI(4)P levels in the Golgi membranes and its redistribution on endosomes (*Dong et al., 2016*). In yeast, depleting VAP orthologs results in the accumulation of PI(4)P at the plasma membrane (*Stefan et al., 2011*).

Altogether, these observations prompted us to hypothesize that VAPs, by regulating PInst homeostasis, could contribute to cell motility processes. Importantly, two partners of VAPs at ER-PM contact sites, ORP3 and Nir2, have been shown to regulate cell motility processes through different pathways (*D'Souza et al., 2020*; *Keinan et al., 2014*; *Lehto et al., 2008*; *Weber-Boyvat et al., 2015*). As a first-line approach, we decided to investigate the effects of VAPA depletion alone on the motility of human adenocarcinoma Caco2 cells. We observed strong motility defects, suggesting that VAPA is required for proper cell motility and that VAPB is not sufficient to compensate for the loss of VAPA. We then aimed to understand the precise function of VAPA in this process. Here, we show that the depletion of VAPA strongly impacts the organization of the actin cytoskeleton, the dynamics of protrusions and the turnover of FA during migration. The role of VAPA stands mainly at the PM where it regulates PI(4)P and PI(4,5)P2 homeostasis, the stability of ER-PM contact sites and their local anchoring at FA, thereby regulating FA disassembly.

## Results

### VAPA is required for proper cell migration and cell spreading

To assess the role of VAPA during cell motility, we generated, using CRISPR/Cas9 gene editing, stable Caco-2 cells either depleted for VAPA (VAPA KO), or expressing non-targeting sequences (Control). VAPA KO cells exhibited a complete and specific depletion of VAPA protein (*Figure 1A and B*). In VAPA KO cells, VAPB was well distributed throughout the ER and its levels were slightly increased, albeit non significant (*Figure 1—figure supplement 1*). We first assessed the capacity of VAPA KO and

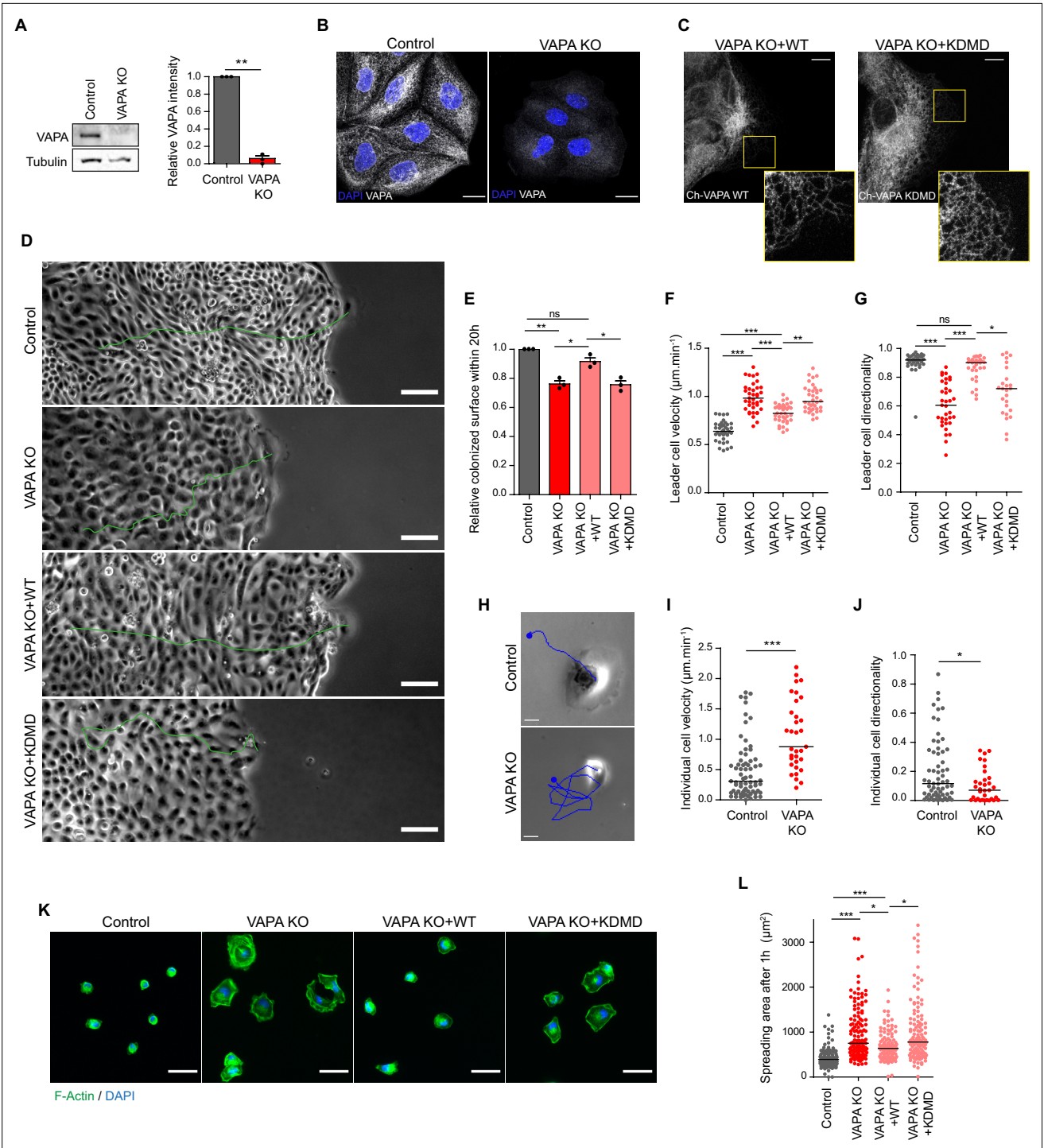

**Figure 1.** VAPA is required for proper cell migration and cell spreading. (**A**) Left panel: Representative immunoblots showing the levels of VAPA and Tubulin in Control and VAPA KO cells. Tubulin expression level was used as loading control. Right panel: quantification of relative VAPA density in Control and VAPA KO cells normalized to Tubulin levels (mean ± SEM from three independent experiments). Data were analysed using a single-sample Student t-test. (**B**) Confocal images of Control and VAPA KO cells immunostained for VAPA. Scale bar 20 μm. (**C**) Confocal images of VAPA KO leader cells expressing wild-type mCherry-VAPA (VAPA KO +WT) or mCherry-VAPA KD/MD (VAPA KO +KDMD). Scale bar 10 μm. (**D**) Phase contrast images of Control, VAPA KO, VAPA KO +WT and VAPA KO +KDMD cells migrating collectively 48 hr after space release. For each condition, the trajectory of a leader cell over 20 hr is shown in green. Scale bar: 100 μm. (**E**) Analysis of relative colonized surface within the last 20 h by Control, VAPA KO, VAPA KO +WT and VAPA KO +KDMD cells (mean ± SEM from three independent experiments). Data were analysed using a One-way Anova paired test. (**F, G**) Analysis of Control, VAPA KO, VAPA KO +WT and VAPA KO +KDMD leader cell velocity (**F**) n=40, 40, 3,9 and 39 cells respectively from three

*Figure 1 continued on next page*

*Figure 1 continued*

independent experiments and directionality coefficient (**G**) n=34, 36, 32, and 26 cells respectively from three independent experiments. Data were analysed using a One-way Anova Kruskal-Wallis test. (**H**) Phase contrast images of a Control and VAPA KO individual cell displacing on fibronectin-coated glass during 5 minutes. The cell trajectory is shown in blue. Scale bar: 20 µm. (**I, J**) Analysis of cell velocity (**I**) and directionality coefficient (**J**) of Control and VAPA KO individual cells displacing on fibronectin-coated glass during at least 3 h (mean ± SEM; Control: n=64 cells; VAPA KO: n=33 cells from two independent experiments). data were analysed using non parametric Mann-Whitney t-test. (**K**) Epifluorescence images of Control, VAPA KO, VAPA KO +WT and VAPA KO +KDMD cells after 1 hour spreading on fibronectin-coated glass and stained as indicated. Scale bar: 50 µm. (**L**) Analysis of cells area after 1 hour spreading on fibronectin-coated glass (mean ± SEM; n=117, 134, 113, and 128 cells respectively from three independent experiments). Data were analysed using a One-way Anova Kruskal-Wallis test. (ns: non significant, ***p-values <0.001, **p-values <0.01, *p-values <0.05).

The online version of this article includes the following source data and figure supplement(s) for figure 1:

**Source data 1.** Table containing the raw data used for the quantifications in *Figure 1A, E, F, G, I, J and L*.

**Source data 2.** Folder containing the 2 original files of the full raw unedited blots for VAPA and Tubulin presented in *Figure 1A and a* figure with the uncropped annotated blots.

**Figure supplement 1.** Analysis of VAPB expression in VAPA KO cells.

**Figure supplement 1—source data 1.** Table containing the raw data used for the quantifications *Figure 1—figure supplement 1A*.

**Figure supplement 1—source data 2.** Folder containing the 2 original files of the full raw unedited blots for VAPB and Tubulin presented in *Figure 1—figure supplement 1A* and a figure with the uncropped annotated blots.

Control monolayers to migrate collectively upon space release on fibronectin-coated glass. In order to avoid any proliferation bias, we treated the cells with mitomycin. In these conditions, VAPA KO monolayers filled the open space less efficiently than Control monolayers (*Figure 1D and E*, *Video 1*). The analysis of leader cells trajectories revealed that VAPA KO leader cells were displacing faster but with a remarkable lack of directionality compared to Control leader cells (*Figure 1F and G*), which could account for the slower colonization capacity VAPA KO monolayers. This collective migration impairment was at least partially cell-intrinsic, as single VAPA KO cells displacing on a 2D surface exhibited identical defects compared to VAPA KO leader cells in monolayers (*Figure 1H–J*). Moreover, upon plating, VAPA KO cells were spreading much faster than Control cells, also indicating a cell-autonomous defect arising in these cells (*Figure 1K and L*). In order to determine the contribution of the MSP domain of VAPA in these phenotypes, we generated stable VAPA KO cell lines expressing a Wild-type form of VAPA (named VAPA KO +WT) or an MSP-mutated form of VAPA (named VAPA KO +KDMD) bearing the K94D M96D mutation that was previously shown to abolish FFAT binding (*Kaiser et al., 2005*). Both VAPA constructs, fused to mCherry, localized to the ER (*Figure 1C*). The expression of mCherry-VAPA WT restored either partially or totally the collective migration capacity of VAPA KO cells (*Figure 1D–G* and *Video 1*) and their spreading behaviour (*Figure 1K and L*), indicating that the loss of VAPA was indeed responsible for the defects observed in VAPA KO cells. Importantly, the VAPA KDMD mutant failed to rescue these phenotypes in VAPA KO cells (*Figure 1D–G , and K–L* and *Video 1*). Altogether, our results identify VAPA as a novel actor of collective and individual cell motility processes, and point to a determinant role of the FFAT-binding MSP domain in VAPA protein.

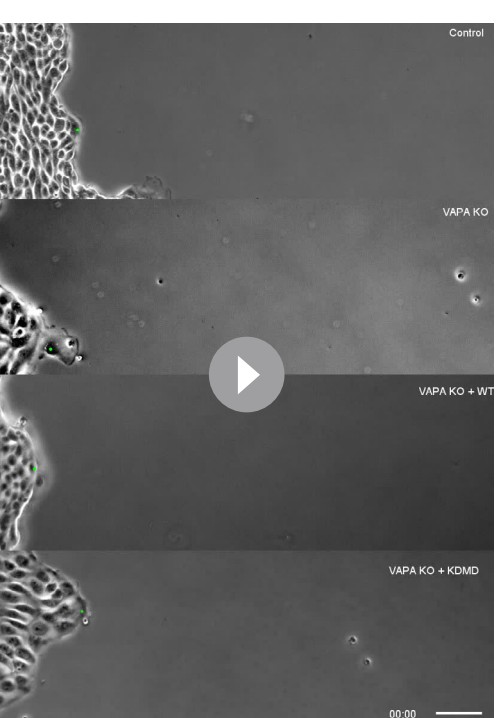

**Video 1.** Phase contrast movie showing collective migration behaviour of Control cells, VAPA KO cells, VAPA KO cells expresing Cherry-VAPA WT (VAPA KO +WT) or Cherry-VAPA KDMD (VAPA KO +KDMD) 48 hr after space release. Scale bar: 100 µm. Time stamp is hour:min.

https://elifesciences.org/articles/85962/figures#video1

## VAPA regulates focal adhesions and actin cytoskeleton through its MSP domain

In order to characterize the motility defects of VAPA KO cells, we studied the organization of cell-matrix adhesions and actin cytoskeleton in these cells. To facilitate the analysis, we decided to focus on leader cells, standing at the leading edge of migrating monolayers (24–48 hr after insert removal), as these cells move in a given direction with a polarized and reproducible morphology. We observed an alteration of cell-matrix adhesions in VAPA KO cells that formed larger central FA compared to Control cells (*Figure 2A and B*). Peripheral FA, located at the leading edge tip where nascent adhesions form, exhibited the same size in Control and VAPA KO cells (*Figure 2A and C*). The expression of VAPA WT in VAPA KO cells restored the size of central FA, but it was not the case with VAPA KDMD, indicating that VAPA regulates FA size through its association with proteins containing an FFAT motif (*Figure 2A–C*). We then compared the organization of actin cytoskeleton in the four cell lines. In Control cells, the F-actin and cortactin stainings revealed the presence of numerous and thick parallel transverse actin arcs at the front, accompanied by cortactin-rich subdomains of branched actin standing at the front edge corresponding to lamellipodial extensions (*Figure 2D and E*). We indeed observed periodic waves of protrusion-retraction arising in these cells every 15 min in average (*Figure 2E and H*). By contrast, VAPA KO cells exhibited disorganized and thinner transverse actin arcs, wider cortactin-rich subdomains occupying most of the leading edge (*Figure 2D and F–G*) and giving rise to long-lasting protrusions (*Figure 2E and H*). Similarly to the FA phenotype, we were able to recover the organization of actin, the size of cortactin-rich domains and the frequency of protrusions by expressing VAPA WT but not the MSP mutant VAPA KDMD (*Figure 2D and F–H*). To conclude, we show that VAPA controls different aspects of cell motility, namely the maintenance of FA size and the preservation of the spatial and temporal organization of the actin cytoskeleton. These functions depend on the association of VAPA with FFAT-motif containing proteins such as lipid transfer proteins, suggesting that lipid transfer is probably required.

## VAPA controls the levels of PI(4)P and PI(4,5)P2 at the PM, but is not essential for PI(4)P homeostasis in the Golgi and endosomal compartments

We then intended to decipher the mechanisms by which VAPA could regulate FA size, the actin cytoskeleton and cell motility. To test the lipid transfer hypothesis, we first characterized to which extend its deletion affected PInst homeostasis. Previous studies have shown that depleting both VAPA and VAPB affected the levels of PI(4)P at the Golgi and PM (*Mao et al., 2019*; *Stefan et al., 2011*), and induced the accumulation of PI(4)P in early endosomes (*Dong et al., 2016*). Using a GFP-PH-OSBP probe, we detected a major intra-cellular pool of PI(4)P at similar levels in the Golgi compartment in Control and VAPA KO cells (*Figure 3A–C*). Importantly, the depletion of VAPA alone was not sufficient to observe an accumulation of PI(4)P in early endosomes (*Figure 3B and C*) which is in agreement with the SupFig. S1D by *Dong et al., 2016* showing no effect on intra-cellular PI(4)P distribution when only VAPA is depleted in HeLa cells. This result suggests that VAPB, which is slightly more abondant in VAPA KO cells, can compensate for the absence of VAPA at ER-Golgi and ER-endosome contact sites thus maintaining PI(4)P homeostasis at the Golgi and endosomes in VAPA KO Caco-2 cells. We then analyzed the amount of PI(4)P and PI(4,5)P2 at the PM using the mCherry-P4M SidM (*Hammond et al., 2014*) and the RFP-PH-PLCδ1 probes respectively. The depletion of VAPA induced a decrease of the PM/cytosol ratio for both PI(4)P and PI(4,5)P2 in protrusive sub-domains of the leading edge (*Figure 3D, E, H and I*). The decrease of PI(4,5)P2 in VAPA KO cells was also observed at the dorsal side of the PM by immunofluorescence staining with anti-PI(4,5)P2 antibodies (*Figure 3F and G*). Altogether, these results show that in Caco-2 cells, VAPA is required to maintain a certain level of PI(4)P and PI(4,5)P2 specifically at the plasma membrane, but is dispensable for PI(4)P homeostasis at the Golgi and endosomal compartments.

## VAPA stabilizes ventral ER-PM contact sites at the front of migrating cells

Based on our observation pointing to a specific role of VAPA in PInst homeostasis at the PM, we questioned the spatio-temporal distribution of ER-PM contacts sites during cell migration and the role

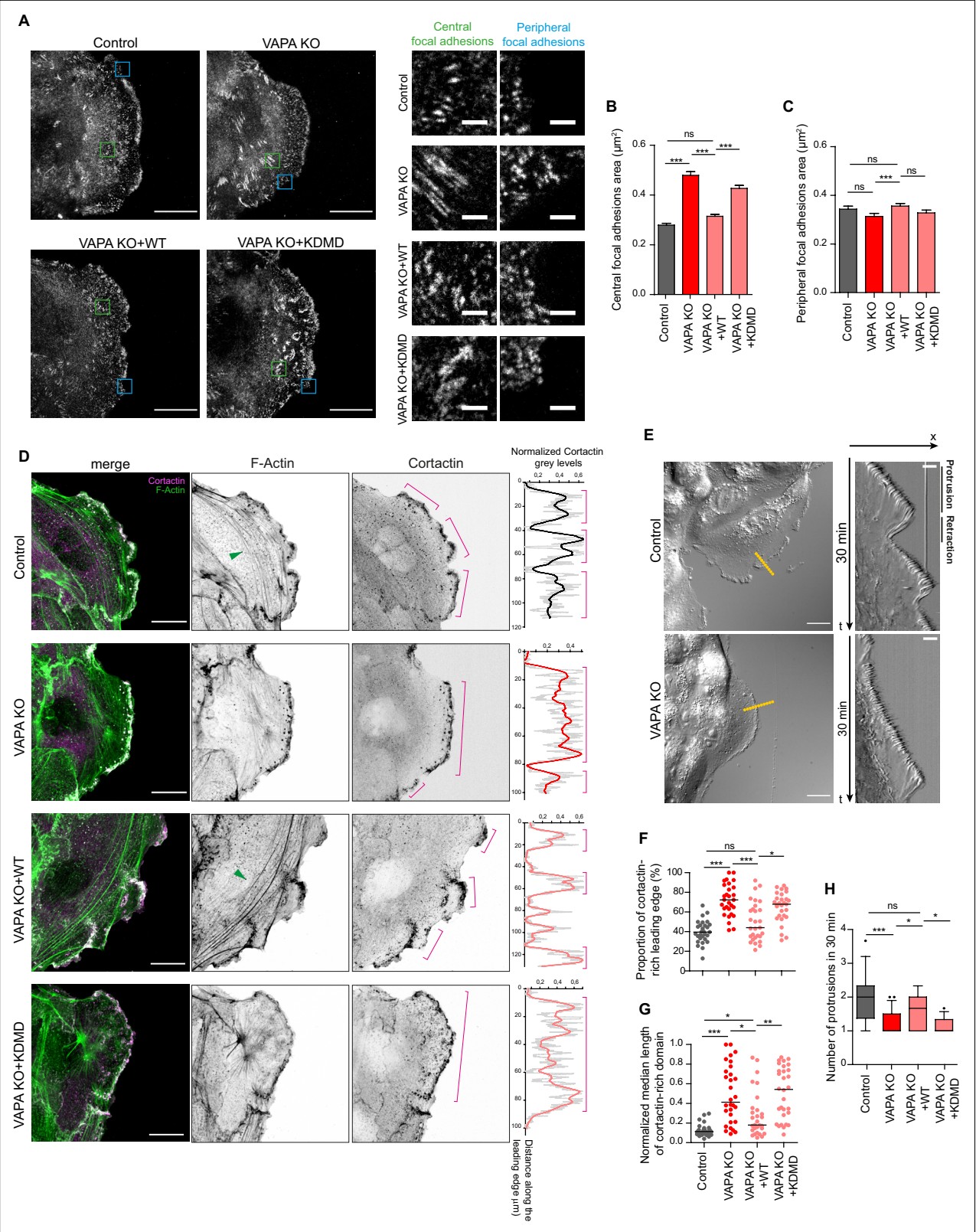

**Figure 2.** VAPA regulates focal adhesions and actin cytoskeleton through its MSP domain. (**A**) Confocal images with zoom boxes of central (green squares) and peripheral (blue squares) paxillin-labeled focal adhesions from Control, VAPA KO, VAPA KO +WT and VAPA KO +KDMD leader cells in migrating monolayers 24 -48hr after insert removal. Scale bar: 20 µm (2 µm in insets). (**B, C**) Analysis of central (**B**) and peripheral (**C**) focal adhesion area quantified from images in A, in Control, VAPA KO, VAPA KO +WT and VAPA KO +KDMD leader cells (Control: n=2968 central and 353 peripheral

*Figure 2 continued on next page*

*Figure 2 continued*

focal adhesions from 20 cells, VAPA KO: n=4329 central and 293 peripheral focal adhesions from 22 cells, VAPA KO +WT: n=3818 central and 336 peripheral focal adhesions from 24 cells, VAPA KO +KDMD: n=2801 central and 297 peripheral focal adhesions from 19 cells, from three independent experiments). Data were analysed using a One-way Anova Kruskal-Wallis test.( D) Confocal images of actin cytoskeleton network in Control, VAPA KO, VAPA KO +WT and VAPA KO +KDMD leader cells stained for cortactin and F-actin. Green arrows point to F-Actin transversal arcs. Plot profiles of the cortactin signal along the leading edge are shown on the right. Pink lines highlight the cortactin-rich protrusive subdomains. Scale bar: 20 μm.( E) Differential Interference Contrast (DIC) images (left) and kymographs (right) along the yellow lines of Control and VAPA KO leader cells from a 30 min movie at 1 frame every 3 seconds, showing protrusion and retraction phases of the leading edge. Scale bar: 20 μm (left) and 5 μm (right). (F, G) Analysis of proportion of the leading edge enriched with cortactin (F) and mean size of cortactin-enriched domains at the leading edge normalized to the length of the leading edge (G) quantified from images in D (mean ± SEM; n=29 cells for each cell line, from three independent experiments). Data were analysed using a One-way Anova Kruskal-Wallis test. (H) Quantification of protrusion phases frequency per 30 min quantified from the kymographs in E, in Control, VAPA KO, VAPA KO +WT and VAPA KO +KDMD leader cells (mean ± SEM; n=16, 21, 17 and 12 cells respectively, from three independent experiments). Data were analysed using a non parametric Mann-Whitney t-test. (ns: non significant, ***p-values <0.001, **p-values <0.01, *p-values <0.05).

The online version of this article includes the following source data for figure 2:

**Source data 1.** Table containing the raw data used for the quantifications *Figure 2B, C, D, F, G and H*.

of VAPA in this organization. To this aim, we took advantage of a fluorescent probe, GFP-MAPPER, which selectively labels ER-PM contact sites (*Chang et al., 2013*). When expressed in Caco-2 cells, GFP-MAPPER distributed as foci along ER tubules both in Control and VAPA KO cells (*Figure 4A*) and appeared at ER-PM appositions detected by TIRF microscopy (*Figure 4B* and *Video 2*). Moreover, we could observe GFP-MAPPER foci along VAPA-containing ER tubules highlighted with an exogenous mCherry-VAPA WT protein expressed in Control cells (*Figure 4C*) or with an anti-VAPA antibody (*Figure 4—figure supplement 1A*). At the front of leader cells, GFP-MAPPER foci divided in two subpools: the dorsal subpool ongoing forward movement with the migrating cell and the ventral subpool remaining immobile relative to the substrate, suggesting that ventral ER-PM contact sites might be anchored to cell-matrix adhesions (*Figure 4—figure supplement 1B* and *Video 3*). Using electron microscopy, we were indeed able to identify numerous ER-PM contact sites sitting at the ventral leading edge of migrating cells (*Figure 4D*). At the ultrastructural level, the depletion of VAPA had no significant effect on ER-PM contact sites regardless of their location (*Figure 4—figure supplement 1C–E*). However, the dynamic analysis of GFP-MAPPER foci revealed that ventral ER-PM contact sites, which barely moved (400 nm/min on average) and persisted generally more than 10 min in Control cells, were significantly less stable both spatially and temporally in VAPA KO cells (*Figure 4E–G* and *Video 4*). To conclude, our results identify a sub-pool of ER-PM contact sites docked to the substrate at the front of migrating cells, and whose spatio-temporal stability requires VAPA. Together with the alteration of FA size in VAPA KO cells, these results prompted us to hypothesize that VAPA might accomplish a function at FA.

## VAPA promotes microtubule-dependent FA disassembly

To further investigate the role of VAPA in FA turnover, we studied more precisely FA dynamics in Control and VAPA KO cells during cell migration by TIRF microscopy. While the assembly rate of FA was similar in both cell lines, their disassembly rate was slower in VAPA KO cells (*Figure 5A–C*). In addition, VAPA KO FA exhibited a longer lifetime in average, with a majority lasting at least 40 min, instead of 30 min in Control cells (*Figure 5A and D*). The expression of VAPA WT in VAPA KO cells restored the lifetime of FA, unlike VAPA KDMD mutant, indicating that not only the size of FA but also their lifetime depended on the interaction of VAPA with FFAT-containing proteins (*Figure 5D*). One way to disassemble FA is through clathrin-mediated endocytosis of integrins which has been shown to depend on microtubules (*Ezratty et al., 2009*; *Ezratty et al., 2005*). To test whether VAPA-mediated disassembly of FA relied on the microtubule network, we synchronized FA disassembly by nocodazole wash-out experiment in migrating cells and measured the evolution of FA size, as described previously. In Control cells, nocodazole wash-out induced the fast recovery of the microtubule network after 15 min (*Figure 5—figure supplement 1A*) concomitantly to the disassembly of FA attested by a 50% reduction of FA size. However, in VAPA KO cells, FA failed to disassemble upon microtubule recovery (*Figure 5E and F*). These results demonstrate that VAPA is required for microtubule-dependent FA disassembly during migration of Caco2 cells.

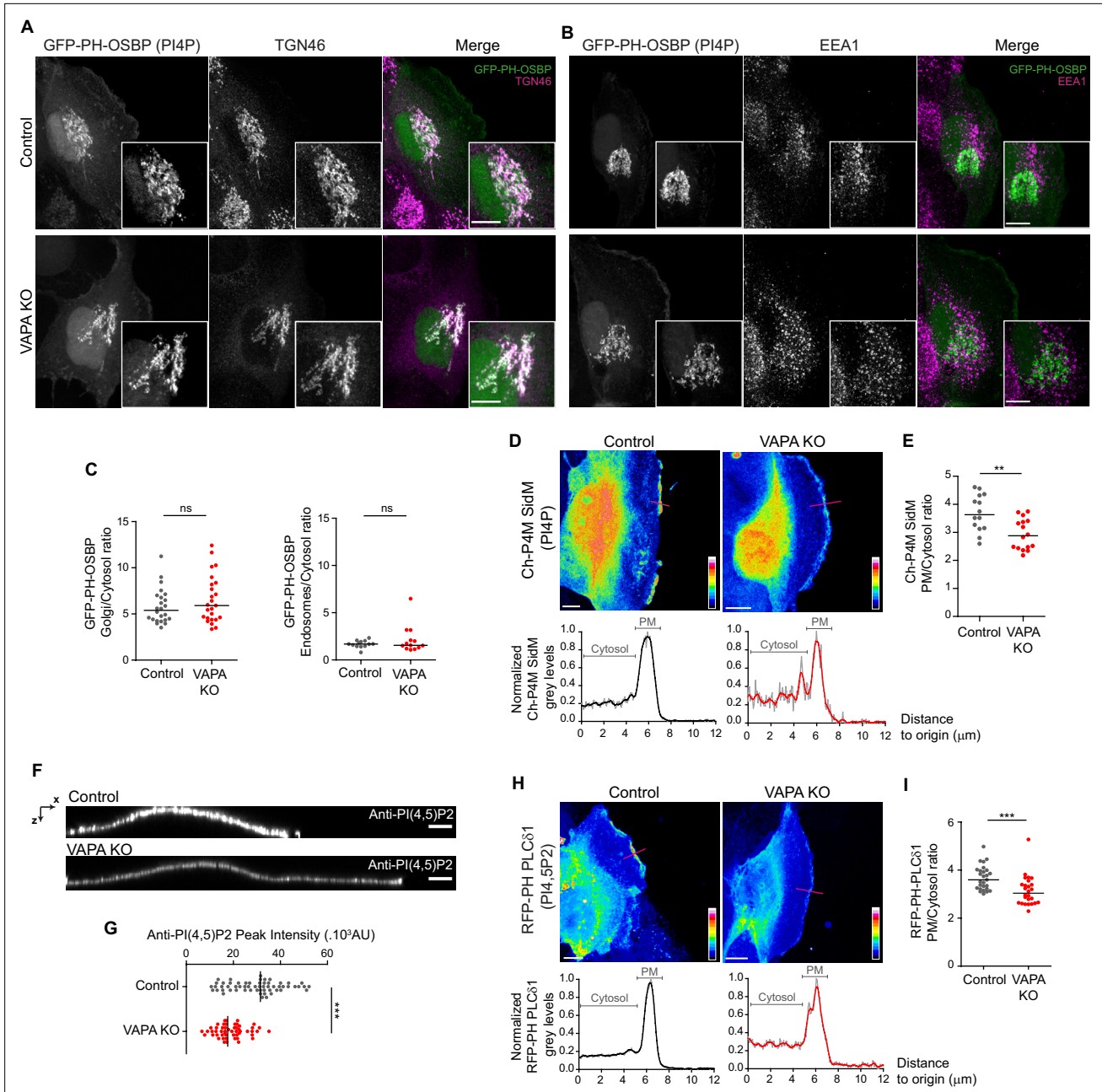

**Figure 3.** VAPA controls PInst homeostasis at the PM, but not in other compartments. (**A, B**) Confocal images and zoom boxes of PI(4)P distribution in Control and VAPA KO leader cells expressing GFP-PH-OSBP and immunostained for TGN46 (**A**) or EEA1 (**B**). Scale Bar: 10 µm. (**C**) Analysis of Golgi(TGN46)/cytosol (left panel) and Early endosomes(EEA1)/Cytosol (right panel) ratio of GFP-PH-OSBP signal, quantified from images in A and B, in Control and VAPA KO leader cells (Left panel: n=24–25 cells, Right panel: n=13 cells; from three independent experiments). (**D, H**) Top: Sum projection of confocal images of PI(4)P (**D**) and PI(4,5)P2 (**H**) distribution in Control and VAPA KO leader cells expressing mCherry-P4M SidM or RFP-PH-PLCδ1 respectively, represented as a color-coded heat map. Scale bar: 10 µm. Bottom: Plot profiles of normalized grey levels along the pink lines. (**E, I**) Analysis of PM/Cytosol ratio of mCherry-P4M SidM (**E**) and RFP-PH-PLCδ1 (**I**), quantified from plot profiles represented in D and H respectively, in protrusive domains at the leading edge of Control and VAPA KO leader cells (E: n=14–16 cells, I: n=26–27 cells; from three independent experiments). (**F**) XZ view of confocal images of Control and VAPA KO cells immunostained for PI(4,5)P2. Scale Bar: 5 µm. (**G**) Analysis of PI(4,5)P2 peak intensity, quantified from images in F, in Control and VAPA KO leader cells (Control: n=58 cells; VAPA KO: n=53 cells, from three independent experiments). All data were analysed using non parametric Mann-Whitney t-test (ns: non significant, ***p-values <0.001, **p-values <0.01, *p-values <0.05).

The online version of this article includes the following source data for figure 3:

**Source data 1.** Table containing the raw data used for the quantifications *Figure 3C, D, E, G, H and I.*

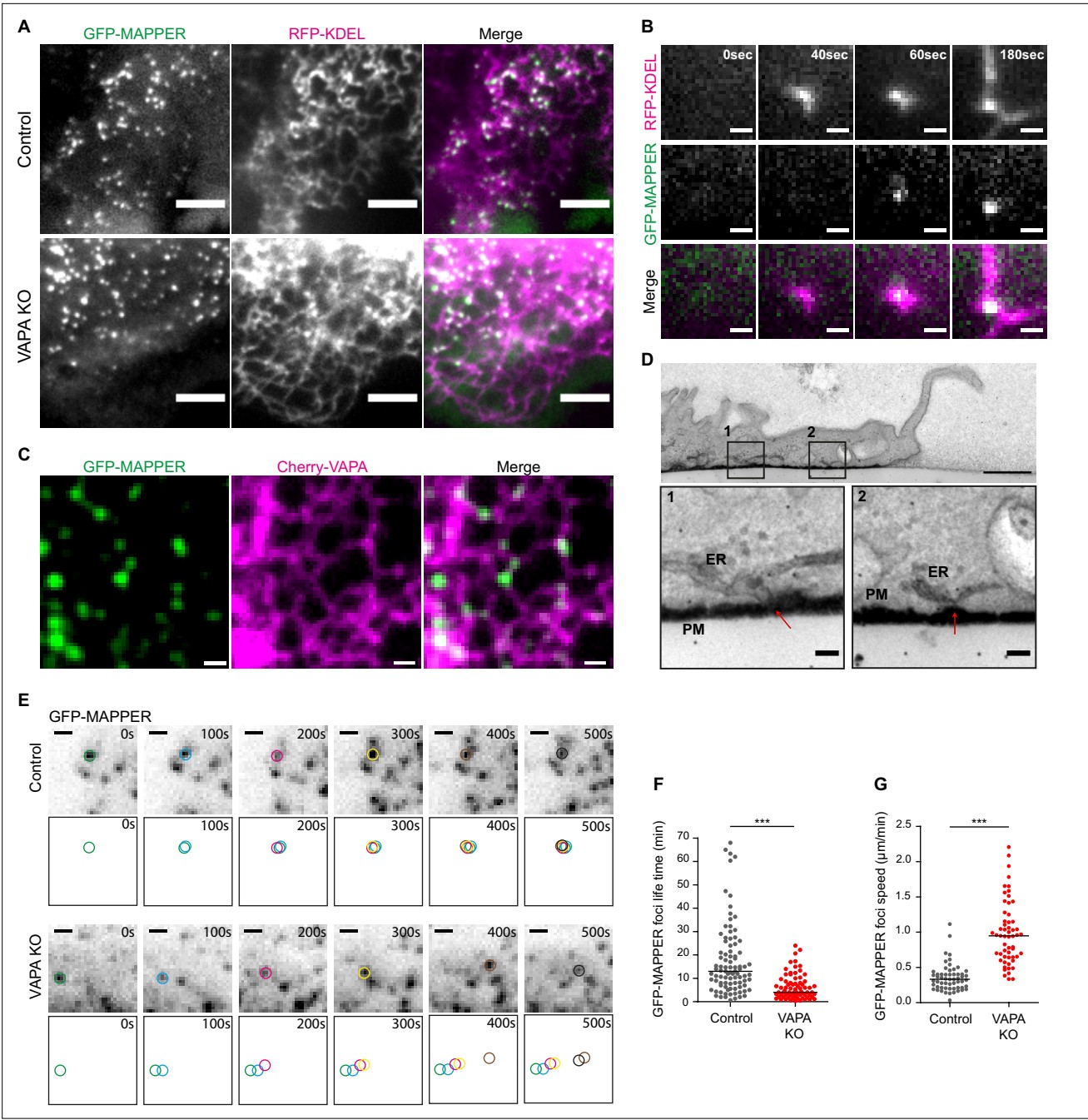

**Figure 4.** VAPA stabilizes ventral ER-PM contact sites at the front of migrating cells. (**A**) TIRF microscopy images of GFP-MAPPER foci distribution along the ER in Control and VAPA KO leader cells expressing GFP-MAPPER and RFP-KDEL. Scale bar: 5 μm. (**B**) Sequential TIRF microscopy images of ER and GFP-MAPPER foci accumulation at the front of a Control leader cell expressing GFP-MAPPER and RFP-KDEL. Scale bar: 1 μm. (**C**) Confocal images of GFP-MAPPER foci distribution along Cherry-VAPA containing ER in Control cells transiently expressing Cherry-VAPA. Scale bar: 1 μm. (**D**) Transmission Electron Microscopy images of transversal cuts of a Control leader cell, showing the leading edge. Arrows point to ER-PM contact sites at the bottom of the cell. Scale bar: 1 μm (top) and 100 nm in insets 1 and 2. (**E**) Sequential TIRF microscopy images of GFP-MAPPER foci at the front of Control and VAPA KO leader cells expressing GFP-MAPPER. Individual GFP-MAPPER foci at each time point are pictured in the frames below images. Scale bar: 1 μm. (**F, G**). Analysis of the lifetime (**F**) and the speed (**G**) of ventral GFP-MAPPER foci, quantified from images in E, in Control and VAPA KO leader cells (Control: n=92 and 52 foci from 8 cells; VAPA KO: n=77 and 58 foci from 6 cells, from four independent experiments). All data were analysed using non parametric Mann-Whitney t-test (***p-values <0.001).

The online version of this article includes the following source data and figure supplement(s) for figure 4:

**Source data 1.** Table containing the raw data used for the quantifications *Figure 4F and G*.

*Figure 4 continued on next page*

*Figure 4 continued*

**Figure supplement 1.** Dynamic and ultrastructural analysis of ER-PM contact sites.

**Figure supplement 1—source data 1.** Table containing the raw data used for the quantifications *Figure 4—figure supplement 1A and D* and 1E.

## VAPA mediates the stable anchoring of ER-PM contact sites to FA before their disassembly

To understand the function of VAPA at FA, we analysed the repartition and the dynamic of ventral ER-PM contact sites relative to FA by TIRF microscopy. In Control cells, the majority of central FA (75%) exhibited one or more GFP-MAPPER foci in their vicinity (*Figure 6A and B*). Using super-resolutive Structured Illumination Microscopy (SIM), we could determine that the peaks of the most proximal GFP-MAPPER foci partially superposed on the peaks of FA, with a distance of around 50–150 nm at 50% of the peaks, suggesting that the two elements were either in contact or in close proximity. Some GFP-MAPPER foci were even found inside the perimeter of FA (*Figure 6—figure supplement 1A*). In VAPA KO cells, only 30% of central FA were standing proximal to at least one GFP-MAPPER foci (*Figure 6A and B*). The expression of VAPA WT or the mutant VAPA KDMD in VAPA KO cells restored the high percentage of central FA proximal to GFP-MAPPER foci (*Figure 6B*), indicating that the MSP domain of VAPA was not required for the physical proximity between central FA and ER-PM contact sites. Unlike central FA, almost none of the peripheral FA were proximal to GFP-MAPPER foci (*Figure 6B*), suggesting that ER-PM contact sites are probably getting closer to FA after their assembly. Indeed**,** when we monitored the time course of mCherry-Vinculin and GFP-MAPPER signals within the perimeter of single FA by TIRF microscopy, we could establish that in Control cells, for most FA (71,4%), the first GFP-MAPPER foci appearing in the FA vicinity was detected after FA assembly and before its disassembly, with a preferential arrival within the 10 min preceding disassembly (*Figure 6C–E*). Once detected, this first GFP-MAPPER foci seemed to anchor to the pre-existing FA, as it was persisting 6 min on average in the FA vicinity (*Figure 6C and F*). In VAPA KO cells, where only 30% of FA were found in proximity to GFP-MAPPER foci (*Figure 6A–B*), there was no preferential time distribution of arrival of the first GFP-MAPPER foci, with 47,6% of them arriving before FA disassembly and 52.4% after FA disassembly (*Figure 6C–E*). Once detected, these foci were rapidly moving away (*Figure 6C and F*). These results show that VAPA is required for the stable anchoring of ER-PM contact sites to pre-existing FA before and at the time of FA disassembly. Imporantly, we were able to detect by TIRF microscopy the presence of VAPA at GFP-MAPPER foci proximal to FA (*Figure 6G and H*) reinforcing the idea that VAPA is indeed playing a role there.

## Discussion

In this work, we identify VAPA as an essential player in cell motility, with implications in different aspects of this process. We show that VAPA is required for both single and collective cell migration, cell spreading, protrusive waves, actin cytoskeleton dynamics and FA turnover. These processes require the MSP domain of VAPA, suggesting that they are regulated through VAPA-mediated lipid transfer mechanisms.

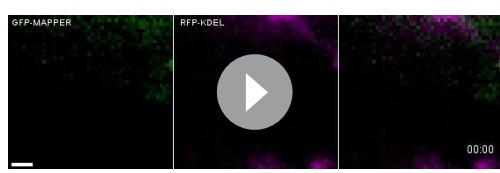

**Video 2.** TIRF microscopy movie showing the accumulation of a GFP-MAPPER foci (green) at a site of close apposition between the ER (RFP-KDEL, Magenta) and the PM, at the front of a Control leader cell. Scale bar: 1 µm. Time stamp is min:sec.

https://elifesciences.org/articles/85962/figures#video2

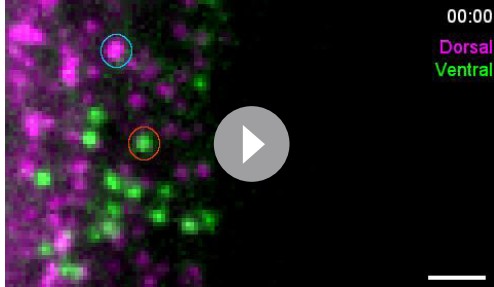

**Video 3.** Confocal microscopy movie showing dorsal (magenta) and ventral (green) GFP-MAPPER foci at the front of a Control leader cell expressing GFP-MAPPER. Circles highlight single GFP-MAPPER foci at the dorsal (blue) and ventral (red) sides. Scale bar: 2 µm. Time stamp is min:s.

https://elifesciences.org/articles/85962/figures#video3

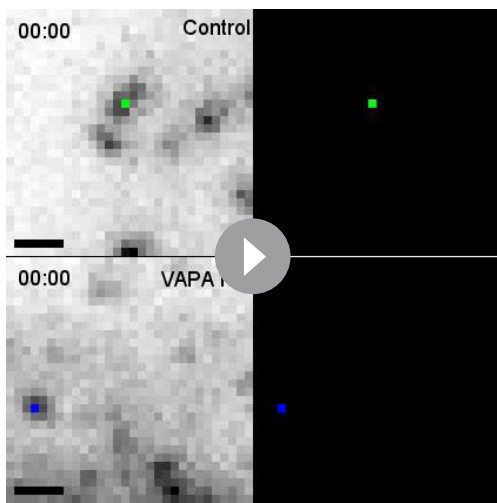

**Video 4.** TIRF microscopy movie showing the dynamics and tracks of GFP-MAPPER foci at the front of Control and VAPA KO leader cells expressing GFP-MAPPER. Scale bar: 1 μm. Time stamp is min:s.

https://elifesciences.org/articles/85962/figures#video4

Our study reveals a non-redundant function for VAPA in cell motility among the VAP family of proteins in Caco2 cells. Indeed, the presence of endogenous VAPB in VAPA KO cells, even at slightly higher levels, was not sufficient to compensate the loss of VAPA. The function of VAPA in cell migration seems to arise from its specific role at ER-PM contact sites where it was necessary to maintain high levels of PI(4)P and PI(4,5)P2 independently of VAPB. It was not the case at the Golgi or endosomes, where the depletion of VAPA alone had no effect on PI(4)P levels, which is in agreement with previous observations by Dong et al in HeLa cells depleted for VAPA only (SupFigS1D of *Dong et al., 2016*). This suggests that VAPB might compensate for the absence of VAPA at ER-Golgi and ER-endosomes contact sites and that migration defects in VAPA KO cells were not a result of a dysfunction of lipid transfer in the secretory pathway. Together with previous studies (*Dong et al., 2016*; *Mao et al., 2019*; *Peretti et al., 2008*; *Stefan et al., 2011*), our results converge to the idea that VAPA and VAPB might fulfill both redundant and specific tasks in lipid homeostasis and cellular functions.

PI(4,5)P2 and its phosphorylation product PI(3,4,5)P3 have been described for decades as central orchestrators of cell motility through regulation of actin dynamics, FA maturation and turnover, membrane organization and curvature, and intracellular trafficking (*Mandal, 2020*; *Tsujita and Itoh, 2015*). Notably, several previous studies have shown that altering the global synthesis of PI(4,5)P2, either by knockdown of certain isoforms of the PIPKI kinase which phosphorylates PI(4)P (*Chao et al., 2010a*, *Chao et al., 2010b*; *Thapa et al., 2012*), or by optogenetic approaches (*Idevall-Hagren et al., 2012*; *Xiong et al., 2016*), results in defects that are comparable to the ones observed in VAPA KO cells – namely disorganization of actin, protrusion defects, FA persistence and loss of directional movement. Thus, the phenotypes of VAPA KO cells could be solely explained by a global reduction in PI(4,5)P2 levels at the PM. However, our results clearly show that VAPA not only regulates PI(4,5)P2 homeostasis at the PM, but also fulfills a function at FA where it stably anchors ER-PM contact sites and promotes microtubule-dependent FA disassembly. Importantly, we provide for the first time a quantitative spatio-temporal connection between ER-PM contact sites and FA dynamics in polarized and migrating cells.

## Ideas and speculation

Based on these results, we propose a hypothetical model in which VAPA, through local regulation of lipid transfer in the vicinity of FA would induce the internalization of integrins through clathrin-mediated endocytosis which depends on PI(4,5)P2 levels (*Chao et al., 2010a*, *Chao et al., 2010b*; *Nader et al., 2016*) and on microtubule polymerization (*Ezratty et al., 2009*). Interestingly, MCS have already been shown to directly regulate endocytosis in yeast (*Encinar Del Dedo et al., 2017*).

Through which mechanisms could VAPA control the dynamics of FA and their close proximity with ER-PM contact sites? Our results show that the MSP domain of VAPA is required for the regulation of FA dynamics. Importantly, two lipid transfer proteins interacting with VAPA at ER-PM contact sites, Nir2 and ORP3, have been previously described as regulators of FA dynamics and cell motility, establishing them as potential drivers for the specific function of VAPA at FA. Nir2, which interacts with VAPs at ER-PM and ER-Golgi contact sites and regulates PI transport (*Chang and Liou, 2015*; *Kim et al., 2013*; *Peretti et al., 2008*), favors epithelial-mesenchymal transition and tumor metastasis (*Keinan et al., 2014*). Similarly to VAPA KO cells, the depletion of Nir2 in HeLa cells induces a decrease of PI(4,5)P2 levels at the PM (*Kim et al., 2013*), indicating that a part of VAPA effect on PInst homeostasis could be supported by VAPA/Nir2-mediated lipid transfer. ORP3, which extracts PI(4)P from the PM

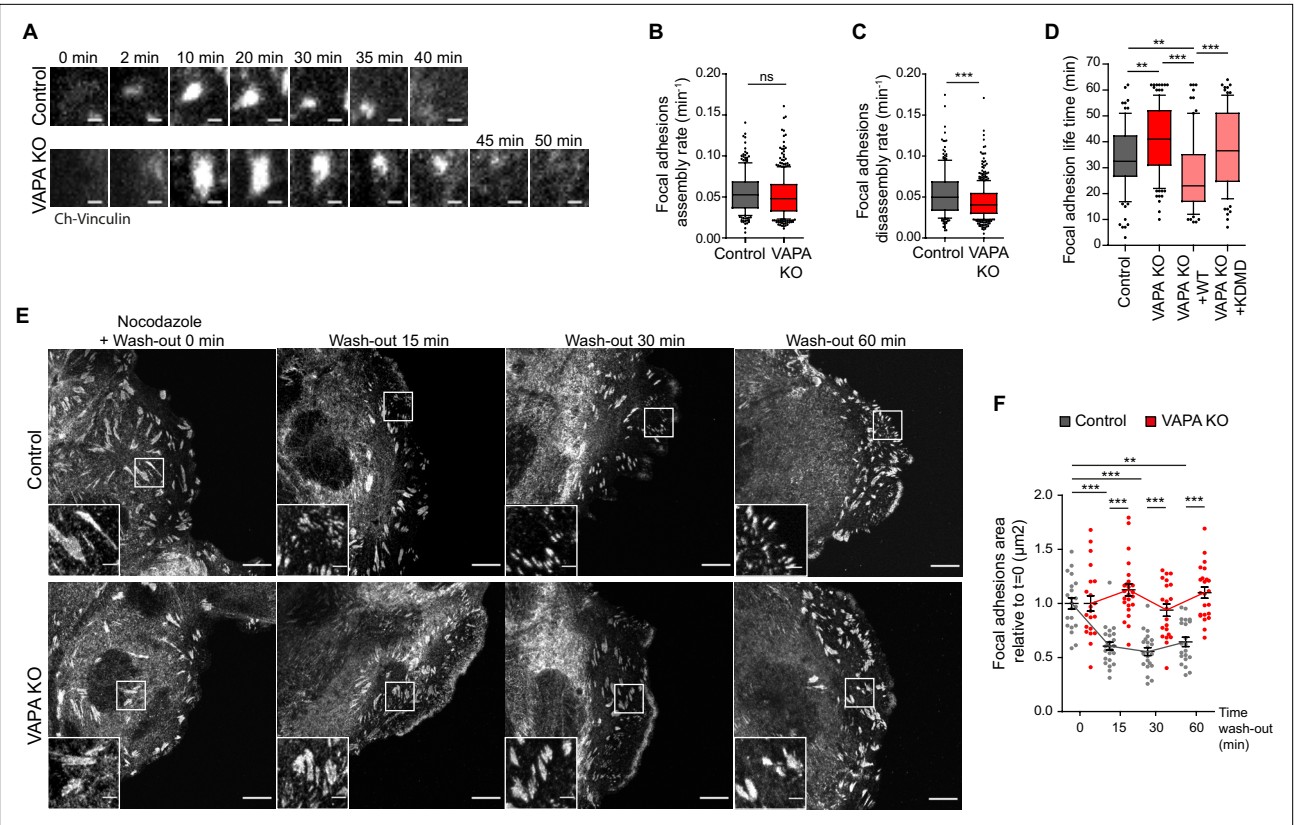

**Figure 5.** VAPA promotes microtubule-dependent FA disassembly. (**A**) Sequential TIRF microscopy images of focal adhesions in Control and VAPA KO leader cells expressing mCherry-Vinculin. Scale Bar: 1 µm. (**B, C**) Analysis of assembly rate (**B**) and disassembly rate (**C**) of focal adhesions (FA), quantified from time-lapse images in A, in Control and VAPA KO leader cells (whisker plots with 10–90 percentile; Control: n=217 FA from 9 cells; VAPA KO: n=342 FA from 8 cells, from three independent experiments). Data were analysed using non parametric Mann-Whitney t-test. (**D**) Distribution of focal adhesion life times in Control, VAPA KO, VAPA KO +WT and VAPA KO +KDMD leader cells (Control: n=98 FA from 10 cells; VAPA KO: n=120 FA from 12 cells, VAPA KO +WT: n=79 FA from 8 cells, VAPA KO +KDMD: n=90 FA from 9 cells, from three independent experiments). Data were analysed using a One-way Anova Kruskal-Wallis test. (**E**) Confocal images of focal adhesions after nocodazole treatment and wash-out in migrating Control and VAPA KO leader cells immunostained for paxillin. Scale bar: 10 µm (2 µm in insets). (**F**) Analysis of relative focal adhesions (FA) size, quantified from images in E, in Control and VAPA KO leader cells after 0 min, 15 min, 30 min, and 60 min after nocodazole wash-out (FA from 22 to 25 cells were analysed, from 3 independent experiments. Control T0min: n=6053 FA; Control T15 min: n=5146 FA; Control T30 min: n=5543 FA; Control T60 min: n=3913 FA; VAPA KO T0 min: n=4481 FA, VAPA KO T15 min: n=3878 FA, VAPA KO T30 min: n=4165 FA; T60 min: n=4889 FA). Data were analysed using non parametric Mann-Whitney t-test. (ns: non significant, \*\*\*p-values <0.001, \*\*p-values <0.01, \*p-values <0.05).

The online version of this article includes the following source data and figure supplement(s) for figure 5:

**Source data 1.** Table containing the raw data used for the quantifications *Figure 5B, C, D and F*.

**Figure supplement 1.** Organisation of microtubules after nocodazole treatment and wash-out.

at ER-PM contact sites (*Gulyás et al., 2020*), positively regulates the R-Ras pathway promoting cell-matrix adhesion (*Lehto et al., 2008*; *Weber-Boyvat et al., 2015*). More recently, ORP3 was found to be recruited to disassembling FA upon calcium influx and to favour their turnover through lipid exchange and crosstalk with STIM1/Orai1-mediated calcium influx (*D'Souza et al., 2020*).

But VAPA might not act at FA just through its MSP domain. Indeed, we show that VAPA favours the proximity between FA and ER-PM contact sites independently of its MSP domain. Thus, we could imagine that VAPA also associates to FFAT-lacking proteins that would mediate the anchoring of ER to FA. Previous studies have highlighted the role of ER proteins in FA turnover and maturation that could be potential candidates. Kinectin, an integral ER membrane protein interacting with kinesin, has been shown to control the apposition of ER to FA through a microtubule-dependent transport (*Guadagno et al., 2020*; *Ng et al., 2016*). Another protein, the phosphatase PTP1B, which localizes to the cytoplasmic face of the ER and was identified as a partner of VAPA using a BioID approach (*Antonicka et al., 2020*), is targeted to newly formed FA and contributes to their maturation through

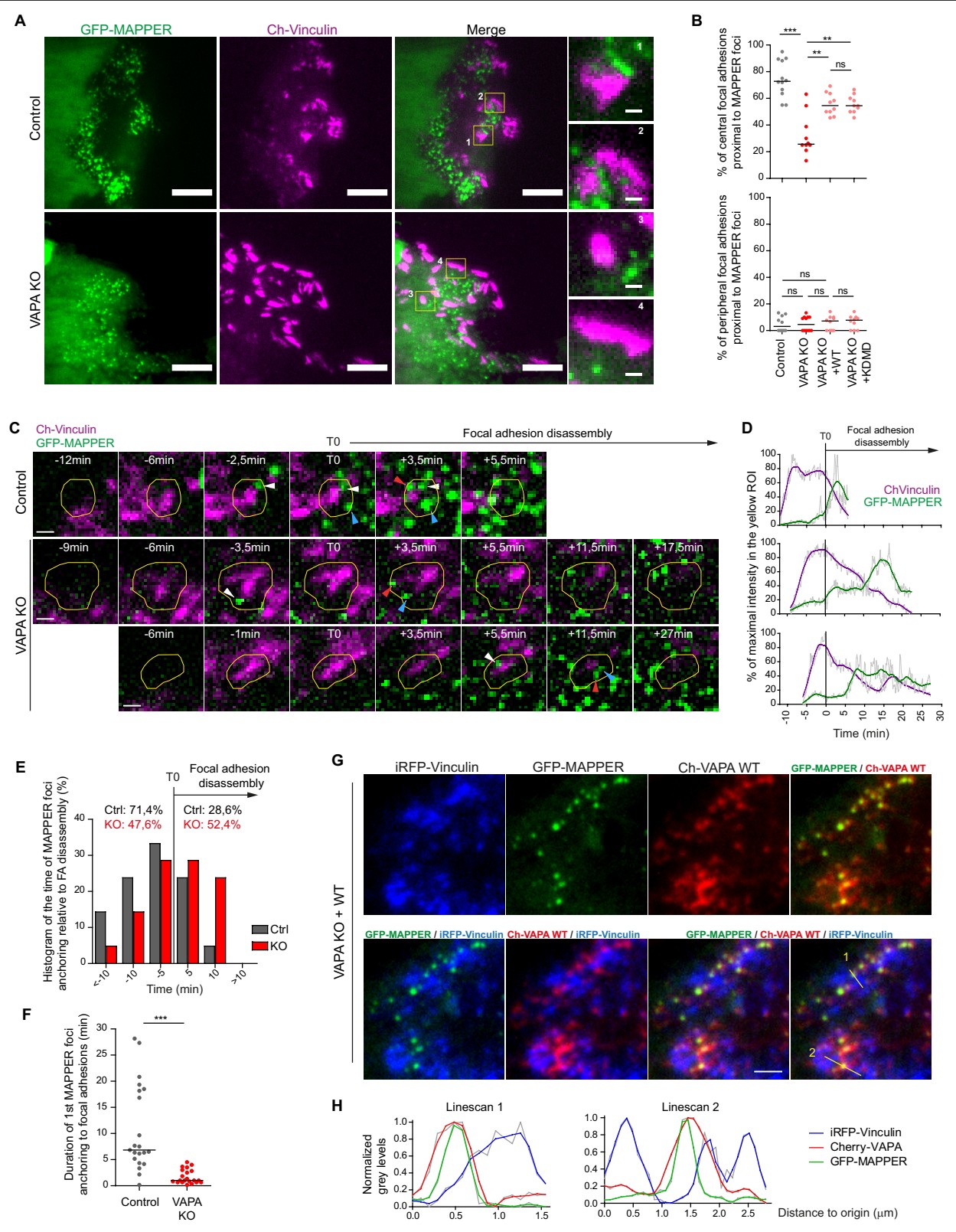

**Figure 6.** VAPA mediates the stable anchoring of ER-PM contact sites to FA before their disassembly. (**A**) TIRF microscopy images of GFP-MAPPER and focal adhesions in Control and VAPA KO leader cells expressing GFP-MAPPER and mCherry-Vinculin. Scale bars: 10 µm (1 µm in insets). (**B**) Analysis of the percentage of central (top) or peripheral (bottom) focal adhesions in contact with GFP-MAPPER foci in Control, VAPA KO, VAPA KO +WT and VAPA KO +KDMD leader cells (n=10–12 cells, from three independent experiments). Data were analysed using a One-way Anova Kruskal-Wallis test.

*Figure 6 continued on next page*

*Figure 6 continued*

(**C**) Sequential TIRF microscopy images of GFP-MAPPER foci and focal adhesions before and after its disassembly in a Control and 2 representative VAPA KO leader cells expressing GFP-MAPPER and mCherry-Vinculin. Scale bar: 1 µm. (**D**) Time course of GFP-MAPPER (green) and mCherry-Vinculin (magenta) signals during the lifetime of focal adhesions in the yellow ROI depicted in C. The signals were smoothed, readjusted to the minimal value and expressed as % of the maximal value. (**E**) Histogram representing the repartition of first anchoring time of GFP-MAPPER foci relative to focal adhesion disassembly in Control and VAPA KO leader cells, quantified from images in D (n=21 focal adhesions from 7 cells from three independent experiments). (**F**) Analysis of duration of the first anchoring of GFP-MAPPER foci to focal adhesions in Control and VAPA KO leader cells, from images in D (n=21 focal adhesions from 7 cells from three independent experiments). Data were analysed using a non parametric Mann-Whitney t-test. (**G**) TIRF microscopy images of iRFP-Vinculin, GFP-MAPPER and Cherry-VAPA WT in a VAPA KO leader cell. Scale bar: 2 µm. (**H**) Plot profiles of normalized grey levels along the two lines depicted in G. (ns: non significant, ***p-values <0.001, **p-values <0.01, *p-values <0.05).

The online version of this article includes the following source data and figure supplement(s) for figure 6:

**Source data 1.** Table containing the raw data used for the quantifications *Figure 6B, D, E, F and H*.

**Figure supplement 1.** Analysis of proximity between FA and GFP-MAPPER foci by super-resolution microscopy.

**Figure supplement 1—source data 1.** Table containing the raw data used for the quantifications *Figure 6—figure supplement 1A*.

dephosphorylation of its substrate p130Cas (*Dadke and Chernoff, 2002*; *Hernández et al., 2006*; *Liu et al., 1998*).

Therefore, the role of VAPA-mediated contact sites at FA could not only be the local modulation of lipid transfer, but also the establishment of a common platform mediating the anchoring of ER and microtubules close to FA where several processes, like lipid transfer, calcium exchange and vesicular transport could converge and participate to FA dynamics.

To conclude, our study reveals unknown functions for VAPA in cell motility processes and in local anchoring of ER-PM contact sites to FA. For a long time, we have assumed that PInst regulation at FA will principally be mediated by the interplay of lipid kinases and phosphatases. Our work reinforces the idea that lipid transfer would be an additional pathway for lipids to be controlled at FA and provides evidences for a spatio-temporal connection between MCS and FA mediated by VAPA. Finally, this work opens new roads to decipher the precise molecular determinants of VAPA function and to explore the role of VAPA during cancerogenesis.

## Materials and methods

The complete list of materials is provided in *Supplementary file 1*.

### Cell culture

CaCo-2 cells, originally acquired from ATCC, were kindly provided by Dr. D. Delacour (IBDM, Marseille). No mycoplasma contamination was detected. CaCo-2 CRISPR control and VAPA knock-out cell lines were maintained in culture in DMEM (1 x) GlutMax 5 g/L D-glucose +pyruvate medium (Gibco) supplemented with 20% fetal bovine serum (Gibco), 100 units/mL penicillin, 100 µg/mL streptomycin and 2 µg/Ml (Gibco) puromycin in 5% CO2 at 37 °C.

### Antibodies, reagents, and plasmids

The following primary antibodies were used: mouse anti-EEA1 monoclonal antibody (BD transduction), rabbit anti-VAPB polyclonal antibody (Atlas antibody), mouse anti-VAPA monoclonal antibody and rabbit anti-VAPA polyclonal antibody for the immunoblot (Sigma), rabbit anti-TGN46 polyclonal antibody (Abcam), mouse anti-paxilline monoclonal antibody (Merck), mouse anti-cortactin monoclonal antibody (Merck), mouse anti-tubulin monoclonal antibody (GeneTex), mouse anti-PI(4,5)P2 monoclonal IgM antibody (Echelon). The following secondary antibodies were used: goat anti-mouse IgM Alexa fluor 555 or Alexa fluor 488, goat anti-mouse IgG Alexa fluor 555 or Alexa fluor 488, goat anti-rabbit IgG Alexa fluor 555 or Alexa fluor 488 (Invitrogen).

The following reagents were used: Hu Plasma Fibronectin (Sigma); Methanol-free formaldehyde (Thermofisher); Fluoromont-G with DAPI (Invitrogen); Alexa fluor 647-conjugated Phalloidin (Invitrogen).

The following plasmids were used; pEGFP-MAPPER (long version, containing the two flexible helical linkers, (EAAAR) 4 and (EAAAR) 6, upstream and downstream flanking regions of the FRB, respectively) was a kind gift from Jen Liou (UT Southwestern Medical Center, USA) (*Chang et al.,*

*2013*), pmCherry-Vinculin was a gift from Chinten James Lim (Addgene plasmid # 80024; http://n2t.net/addgene:80024; RRID:Addgene_80024; *Lee et al., 2013*); iRFP-Vinculin was a gift from Mathieu Coppey (Institut Curie, CNRS UMR168) (*Valon et al., 2017*); pEGFP-PH-OSBP was a gift from Marci Scidmore (Addgene plasmid # 49571; http://n2t.net/addgene:49571; RRID:Addgene_49571; *Moorhead et al., 2010*) pmRFP-PH-PLCδ1 was a kind gift from Francesca Giordano (Institute for Integrative Biology of the Cell, France) *Giordano et al., 2013*; mCherry-P4M-SidM was a gift from Tamas Balla (Addgene plasmid # 51471; http://n2t.net/addgene:51471; RRID:Addgene_51471; *Hammond et al., 2014*) pmRFP-KDEL plasmid was a kind gift from Nihal Altan-Bonnet (National Heart Lung and Blood Institute, NIH, USA) *Altan-Bonnet et al., 2006*; pmCherry-VAPA and pmCherry-VAPA KD/MD, obtained from pEGFP-VAPA and pEGFP-VAPA KD/MD, were kind gifts from Fabien Alpy (IGBMC, France) *Alpy et al., 2013*; CRISPR-Cas9-resistant pmCherry-VAPA and pmCherry-VAPA KD/MD plasmids were obtained by mutagenesis of pmCherry-VAPA and pmCherry-VAPA KD/MD plasmids. Briefly, pmCherry-VAPA and pmCherry-VAPA KD/MD plasmids were amplified by PCR (30 cycles of 10 s at 94 °C, 5 s at 55 °C and 6 min at 72 °C) using the forward 5'TAAGACCGAATTCCGGTATCATCG ATCCAGGGTCAACTGTGACTGTTTCAGT 3' and reverse 5' CGGAATTCGGTCTTACGCAATATC GTCGAGGTGCTGTAGTCTTCACTTTGA 3' primers. PCR products were digested by DpnI enzyme for 2 hr at 37 °C (Agilent) and loaded on a 1% Agarose Gel. PCR amplicons were extracted, purified using the NucleoSpin Gel and PCR clean up kit (Macherey-Nagel) and ligated using the NEBuilder HiFi DNA assembly kit (BioLabs).

## Transfection

For each transfection, 0.5 million Caco2 cells were nucleofected with 2 µg of DNA in 100 µL of T solution using the B024 program on Amaxa Nuclefector II machine, as recommended by the manufacturer (Lonza). CaCo-2 transfected cells were then re-suspended in warm culture medium, replaced 24 hr later.

## Production of CRISPR Cas9 cell lines

Cells were transfected as described above with the double nickase CRISPR Cas9 plasmid targeting human VAPA (Santa Cruz) or a double nickase CRISPR Cas9 Control plasmid (Santa Cruz). The following day, cells were incubated with puromycin at 2 µg/mL. After antibiotic selection, single GFP positive cells were sorted by flow cytometry. Clonal cells were maintained in culture with puromycin and the amount of VAPA protein was detected by western blot and immunofluorescence. VAPA KO cell lines expressing exogenous Cherry-VAPA and Cherry-VAPAKDMD were obtained by transfection of VAPA KO cells with CRISPR-Cas9-resistant pmCherry-VAPA and pmCherry-VAPA KD/MD plasmids. Stable cell lines were generated by selection with Geneticin (Gibco) and sorting of Cherry-positive populations by flow cytometry.

## Western blotting

Confluent cells were lysed in 100 mM Tris pH 7.5, 150 mM NaCl, 0.5% NP40, 0.5% triton-X100, 10% glycerol,1X protease inhibitor cocktail (Roche) and 1 X phosphatase inhibitor (Roche) for 20 min at 4 °C. After 15 min centrifugation at 13,000 *g*, solubilized proteins were recovered in the supernatant. Protein concentration was measured using Bradford assay (Bio-Rad). For SDS PAGE, 50 µg protein extracts were loaded in 4–12% Bis-Tris gel (Invitrogen) or poly-acrylamide gels and proteins were transferred overnight at 4 °C on a nitrocellulose membrane using a liquid transfer system (Bio-Rad). Non-specific sites were blocked with 5% non-fat dry milk in PBS 0.1% Tween 20. Primary antibodies were diluted (1/1000 to 1/500) in PBS 0.1% Tween 20 and incubated overnight at 4 °C. After three washes in PBS 0.1% Tween 20, secondary HRP antibodies diluted in PBS 0.1% Tween 20 (1/10,000) were incubated for 1 hr and washed three times with PBS 0.1% Tween 20. Immunocomplexes of interest were detected using Supersignal west femto maximum sensitivity substrate (Thermo Fisher Scientific) and visualized with ChemiDoc chemoluminescence detection system (Bio-Rad). Quantification of Western blots by densitometry was performed using the Gel analyzer plug in from Image J.

## Immunofluorescence

Cells were fixed with pre-warmed 4% formaldehyde in PBS for 15 min at RT and then washed three times with PBS. For anti-tubulin immunostaining, cells were fixed with frozen Methanol for 15 min

at RT. Permeabilization and blocking were performed in 0.05% saponin/0.2% BSA in PBS for 15 min at RT. The primary antibodies diluted in Saponin/BSA buffer were then incubated overnight at 4 °C. After three washes in saponin/BSA buffer, the samples were incubated with secondary antibodies and, when indicated, Alexa-coupled phalloidin to stain F-Actin in the same buffer for 1 hr at RT. The preparations were washed twice in saponin/BSA buffer, once in PBS, and then mounted with the DAPI Fluoromount-G mounting media.

Immunostaining with anti-PI(4,5)P2 antibody was performed as described by *Kim et al., 2013*. Briefly, cells were fixed at 4 °C for 1 hr in 3.7% formaldehyde / 0.1% glutaraldehyde and incubated in PBS containing 0.1 M glycine for 15 min. Permeabilization, blocking and staining were performed as described above, at 4 °C.

### Cell migration assays

Collective migration assays were performed using 2-well silicon inserts (Ibidi). Glass coverslips were coated with fibronectin solution at 20 µg/mL in water, for 1 hr at room temperature. The cover slip surface was washed with sterile water and air-dried. Ibidi inserts were deposited on the fibronectin coated surface and 40,000–50,000 cells were loaded per well. For fluorescence live imaging, cells were directly plated after transfection in the well. After 3–4 hr of incubation, cell division was blocked using mitomycin at 10 µg/mL in CaCo-2 culture medium, for 1 hr. After overnight incubation in a fresh CaCo-2 culture medium, the insert was removed. Experiments were performed 24 hr to 48 hr after insert removal.

For individual cell migration, cell division was blocked using mitomycin at 10 µg/mL in CaCo-2 culture medium, for 1 hr. After overnight incubation in a fresh CaCo-2 culture medium, cells were detached and plated on a glass coverslip coated with 20 µg/mL fibronectin. After 6 hr, cells were imaged for 24 hr.

Collective and individual cell migration assays were performed using a Zeiss Wide-Field Microscope and imaged at 1 frame every 10 min. Cell trajectories were analysed using Manual Tracking from Fiji. Cell directionality was calculated as the ratio between the net displacement and the trajectory length within the last 20 hr of collective migration.

### Cell spreading assay

Cells were incubated for 1 hr on fibronectin-coated coverslips and fixed with formaldehyde 4% solution as described in the immuno labelling section. Coverslips were mounted on slides and samples were imaged using a Zeiss Apotome fluorescence microscope equipped with a 10 x objective. The spreading area was determined based on Phalloidin-segmented ROI in Fiji/ImageJ.

### Nocodazole experiment

The nocodazole wash-out experiment was adapted from *Ezratty et al., 2005*. Briefly, cells were collected and resuspended in starvation medium (DMEM GlutaMax medium containing 1% fetal bovine serum and 1% of penicillin/streptomycin) and plated in the silicon insert, as described in the Migration assay section. Twenty-four hours later, the insert was removed and cells were left migrating in the starvation medium during 24 hr. Cells were then treated during 2 hr with 10 µM nocodazole diluted in starvation medium. Then, nocodazole was washed-out with starvation medium and cells were fixed either immediately ('0 min Wash-out') or after 15, 30, or 60 min wash-out, using 4% formaldehyde. Focal adhesions and microtubules were labelled as described above.

### Electron microscopy

Migrating cells were fixed in a 2% formaldehyde/1% glutaraldehyde in PBS solution for 1 hr at room temperature, then washed in PBS. Cells were embedded in a gel of 10% BSA and 10% gelatin. Post fixation was performed in reduced osmium (1% $OsO_4$ +1% $K_3Fe(CN)_6$) in water at 4 °C for 1 hr. After extensive washes, cells were incubated in 1% Thiocarbohydrazine in water for 20 min at room temperature and then washed and incubated with 2% $OsO_4$ in water for 30 min at room temperature. After washes, cells were contrasted with 1% uranyl acetate overnight at 4 °C. The following day, the uranyl acetate solution was removed and the samples washed using pure water. Cells were incubated with a pH 5.5 lead nitrate in aspartic acid solution for 30 min at room temperature, washed in water and dehydrated in successive ethanol baths. Cells were embedded in Agar Low Viscosity Resin (Agar

Scientific). 70 nm-thick thin sections were cut using a UC6 ultramicrotome from Leica and deposited on EM grids. Electron microscopy acquisitions were performed using a 120kV Tecnai 12 electron microscope (ThermoFisher) equiped with a OneView 4 K camera (Gatan).

## Analysis of protrusive activity

The analysis of protrusive activity was performed on DIC images acquired every 3 s for 30 min. Kymographs representing the evolution of the leading edge in time along a line were generated and analysed with Fiji. Positive slopes were considered as protrusions and negative slopes as retractions. For each cell, the mean time spent in protrusion and the mean frequency of protrusions.was calculated from 2 to 4 kymographs.

## Measurement of focal adhesion assembly and disassembly

Cells were transfected as described above using the mCherry-Vinculine plasmid and migration assays were performed on fibronectin coated 1.5 H Glass bottom Dishes (Ibidi). Cells were imaged using a TIRF microscope at 1 frame every minute for 1–2 h. Focal adhesion assembly and disassembly rates were obtained using the focal adhesion analysis server (https://faas.bme.unc.edu/; *Berginski and Gomez, 2013*). Focal adhesion assembly and disassembly rate tracks were generated. The tracks with a R-squared equal to or greater than 0.8 were used for the analysis.

## Quantification of PI(4)P and PI(4,5)P2 levels

To characterize the intracellular pool of PI(4)P at the Golgi and on endosomes, colocalization between the GFP-PH-OSBP and TGN46 or EEA1 signals were obtained using Fiji Pearson coefficient plugins. The relative levels of PI(4)P in the Golgi or in endosomes were calculated as the ratio between GFP-PH-OSBP signal in the Golgi or endosomes masks and GFP-PH-OSBP signal in the cytosol, using segmentation tools in Fiji.

The PM to cytosol signal ratios of Ch-P4M-SidM and RFP-PH-PLCδ1 were measured along lines of 5–15 microns at the leading edge of migrating cells, in protrusive domains where the signal was the most enriched. The ratio was calculated as the maximal peak intensity along the line divided by the mean intensity value of RFP-PH-PLCδ1 in the cytosol. The plots on the graph represent the mean of 3 ratios at 3 different locations per cell.

For the quantification of endogenous PI(4,5)P2, cells were immnuno-stained with anti-PI(4,5)P2 antibodies. Confocal imaging was performed using the same acquisition parameters between control and VAPA KO cells. For one z-section, three lines were drawn per cell and the mean of maximal fluorescence intensity at the peak was plotted.

## Analysis of ER-PM contact sites on electron microscopy images

Electron microscopy images were analysed using Fiji/ImageJ. For the analysis, only ER-PM appositions of less than 30 nm distance were considered as ER-PM contact sites.

## Analysis of GFP-MAPPER foci dynamics

GFP-MAPPER foci on TIRF images were tracked using the Manual Tracking plugin from Fiji. The speed corresponds to the mean of instantaneous speeds for each foci. The lifetime was determined visually on the basis of the intensity levels and corresponds the time spent between the first visual appearance of a foci and its visual disappearance.

## Quantification of FA in contact with ER-PM contact sites

The quantification of FA in contact with ER-PM contact sites was performed on TIRF microscopy acquisitions. Ten to 30 FA were randomly selected per cell. An FA was considered to be in contact with an ER-PM contact sites if there was at least one pixel recovery between GFP-MAPPER signal and mCherry-vinculin signal.

## Plot profiles quantification

The normalized grey levels represented in the plot profiles were obtained by first subtracting the minimal grey level to all raw grey levels, and then, by normalizing to the maximal grey level.

## Images acquisition and analysis

For live-microscopy experiments, the samples were placed in a chamber equilibrated at 37 °C under 5% $CO_2$ atmosphere.

Confocal live microscopy imaging was acquired using a Yokogawa CSU-X1 Spinning Disk confocal mounted on an inverted motorized Axio Observer Z1 (Zeiss) and equipped with a sCMOS PRIME 95 (photometrics) camera, using a 63 x/1.4 Plan-Apochromat Oil DIC objective.

TIRF and Super resolution 3D Lattice SIM images were acquired on a ELYRA 7 Zeiss microscope equipped with Two cameras Edge 4.2 CLHS (PCO), using 63 x/1,4 Plan-Apochromat Oil DIC M27 objective. Images were acquired and processed with SW ZEN Black 3.4 software.

For fixed samples, images were acquired with a Zeiss Apotome fluorescence microscope equipped with a 63 X oil immersion objective or with a Zeiss LSM 780 confocal microscope equipped with a 63 X/1.4 Plan-Apochromat Oil DIC objective at a resolution of 0.3 µm z-stacks. Image processing and analysis were done on Fiji software.

## Acknowledgements

We thank Delphine Delacour, Fabien Alpy, Simon de Beco and the past and current members of Membrane Dynamics and Intracellular Trafficking team at Institut Jacques Monod for helpful discussions. We acknowledge the ImagoSeine core facility of the IJM, member of the France BioImaging infrastructure (ANR-10-INBS-04) and GIS-IBiSA, with support from La ligue contre le Cancer (R03/75-79), the Region Île-de-France (Sesame), Université Paris Cité (Labex Who am I?, ANR-11-LABX-0071, Idex ANR-11-IDEX-0005–02), Inserm (Plan Cancer), Région Ile de France (SESAME) and Fondation Bettencourt Schueller. This work was supported by grants from Gefluc groupement Les Entreprises Contre le Cancer (M.L.H), Cancéropôle Ile de France and INCa (M.L.H) (2021–1-EMERG-51), La Ligue contre le cancer (M.L.H) (RS23/75-66), Université Paris Cité (M.L.H) (Labex Who am I?, ANR-11-LABX-0071, Idex ANR-11-IDEX-0005–02) and the Agence Nationale de la Recherche (J-M.V) (ANR-20-CE13-0021). H.S was supported by fellowships from La Ligue contre le cancer and the Fondation pour la Recherche Médicale. M.L.H acknowledges René-Marc Mège, Benoît Ladoux and Delphine Delacour for kindly providing antibodies and plasmids during the starting period of this work.

## Additional information

### Funding

| Funder | Grant reference number | Author |
| --- | --- | --- |
| Gefluc Les Entreprises contre le Cancer | | Mélina L Heuzé |
| Canceropôle Ile de France and INCa | 2021-1-EMERG-51 | Mélina L Heuzé |
| La Ligue contre le Cancer | RS23/75-66 | Mélina L Heuzé |
| Université Paris Cité | Labex Who am I ?, ANR-11-LABX-0071, Idex ANR-11-IDEX-0005-02 | Mélina L Heuzé |
| Agence Nationale de la Recherche | ANR-20-CE13-0021 | Jean-Marc Verbavatz |
| La Ligue contre le Cancer | Doctoral Fellowship | Hugo Siegfried |
| La Fondation pour la Recherche Médicale | 4th year fellowship | Hugo Siegfried |

The funders had no role in study design, data collection and interpretation, or the decision to submit the work for publication.

## Author contributions

Hugo Siegfried, Conceptualization, Formal analysis, Validation, Investigation, Visualization, Methodology, Writing – original draft, Writing – review and editing; Georges Farkouh, Agathe Verraes, Investigation; Rémi Le Borgne, Catherine Pioche-Durieu, Formal analysis, Investigation, Methodology; Thaïs De Azevedo Laplace, Lucien Daunas, Formal analysis, Investigation; Jean-Marc Verbavatz, Conceptualization, Supervision, Funding acquisition, Methodology, Writing – original draft, Project administration, Writing – review and editing; Mélina L Heuzé, Conceptualization, Formal analysis, Supervision, Funding acquisition, Validation, Investigation, Visualization, Methodology, Writing – original draft, Project administration, Writing – review and editing

## Author ORCIDs

Catherine Pioche-Durieu ⦿ http://orcid.org/0000-0003-0988-1169
Mélina L Heuzé ⦿ http://orcid.org/0000-0002-4271-2706

## Decision letter and Author response

Decision letter https://doi.org/10.7554/eLife.85962.sa1
Author response https://doi.org/10.7554/eLife.85962.sa2

## Additional files

### Supplementary files

• MDAR checklist

• Supplementary file 1. Complete list of materials used in this study indicating the references and dilutions used for the antibodies.

### Data availability

All data generated or analysed during this study are included in the manuscript and supporting files; Source Data files have been provided for all figures.

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
