## [Editor Report]

This manuscript presents important findings that bring together two important topics in cell biology: the function of membrane contact sites and cell migration. The authors present compelling evidence to support a role of the ER tether protein VAP-A in focal adhesion dynamics and cell motility. This paper will be of interest to those cell biologists and biophysicists working on adhesion, migration, and membrane contact site biology.

---

## [Decision Letter]

**Decision letter after peer review:**

Thank you for submitting your article "The ER tether VAPA is required for proper cell motility and anchors ER-PM contact sites to focal adhesions" for consideration by *eLife*. Your article has been reviewed by 3 peer reviewers, including Felix Campelo as the Reviewing Editor and Reviewer #1, and the evaluation has been overseen by Suzanne Pfeffer as the Senior Editor.

Essential Revisions:

We all agree that this manuscript reports potentially very interesting findings, but we also think that additional data will be needed to support the claims. The essential revisions requested are the following (1-3 will require new experiments, 4-6 might not, but additional discussion is requested):

1) Show additional rescue experiments (and/or test some of the findings with other KO clones). Especially regarding those experiments presented in Figures 1,5, and 6. We all agreed that this is perhaps the major gap in the paper. See detailed reviews below.

2) Show VAP-A localization by IF together with the localization of GFP-MAPPER MCS.

3) Test how peripheral vs. more centrally organized FAs behave with respect to VAP-A.

4) Improve their assessment/discussion on the PI4P and PIP2 results.

5) Improve on the method explanation/statistics description, etc.

6) The model is very speculative with respect to the data presented in the manuscript. This is in principle OK, but then it really needs to be discussed more as a speculative/working hypothesis rather than a summary model. While it would be great to be able to experimentally test some of the model's predictions (e.g., test some of the observed effects by KO Nir2, ORP3, or monitor integrin trafficking), we are not requesting any additional experiments in this regard.

*Reviewer #1 (Recommendations for the authors):*

As mentioned in the public review, I think that this is a very interesting paper, with potentially major findings that will help us understand better the process of cell migration. However, I think that, although parts of the paper are based on solid evidence, there are some important points of the paper that would require additional, stronger, and more compelling evidence.

– The VAP-A KO cell line is produced by single clonal selection. I really appreciate the experiments shown in Figure 2A, where the phenotype in the KO was rescued by the expression of WT and not mutant VAP-A. However, similar rescue experiments are not performed elsewhere in the manuscript. I am of course not saying that the rescues are presented in every single experiment, but in some of the key experiments, additional evidence about the specificity of the VAP-A effect should be provided (e.g. have the authors generated additional clonal KOs and tested the effects in those cell lines?)

– It was not clear to me, based on the image in Figure 2B, that the cortactin signal in the KOs is different from Control. Could the authors quantify that?

– Line 109: the title of the section reads as "VAPA regulates FAs and actin cytoskeleton through its MSP domain". However, the authors have not tested the role of the MSP domain in the actin effects. Hence, a rescue experiment with the WT and mutant VAP-A would be necessary here.

– Figure 6: This is a potentially quite cool finding. However, based only on this I'm not sure that the claims in the paper are sustained by the evidence. For instance, is a similar effect as that presented in panel E seen for those FAs, not in contact with an ER-PM MCS (according to panel B, there are about 25% of those)? Similarly, doing this experiment in the KO cells, where there is a fraction (>30%) of FAs proximal to an MCS, would determine whether MCS are enough for this effect or whether VAP-A is indeed playing a role there.

– The model is interesting and I think it presents a lot of possibilities for future work. However, as it is now, I think it is very hypothetical (which the authors already state in the legends), and mostly based on published work. This is fine, however, I feel that, given some of the cell-specific differences seen in this report (e.g. Golgi PI4P changes), the ideas in the cartoon might not be 100% extrapolated to their system. Of course, it would be great if the authors test the effects of depleting other ER-PM MCS proteins (Nir2, ORP3), or study integrin uptake, etc. However, I think this would be too much to ask for this paper (so it is not necessary in my opinion), but then this section might need to be integrated as an "Ideas and Speculation" section, according to *eLife* editorial: https://elifesciences.org/inside-*eLife*/e3e52a93/*eLife*-latest-including-ideas-and-speculation-in-*eLife*-papers

Other important points:

– Regarding the results in Figure 1C, I, J: The KOs look overall to be larger cells. Is this true? Is this due to clonal selection or because of the KO of VAP-A? Would the authors rescue the phenotype with the re-expression of the WT VAP-A? I think it would be informative to measure the final area/single cell after 24 h of spreading (steady state) and/or measure the nuclear area to be sure these effects are not due to size variability but to VAP-A depletion. Also, images in panel (I) are very pixelated in my PDF.

– The values plotted in Figure 3C are surprisingly small if they are indeed reporting at the fluorescence intensity density in the Golgi region relative to the same quantity measured elsewhere in the cytosol. Could the authors double-check if the quantification is done correctly, or maybe I am missing something?

– Figure 4E: I think the results are clear, but their representation of them could be improved to help the readers. I'd suggest that the tracking performed in the panels below the TIRF frames are showily more data points, and linked through connecting trajectory lines. Also, explain in the methods how this tracking and the lifetime and speed (panels F, G) have been measured.

– This might be something obvious for the authors/other experts in the field, but why is it that VAP-AKO cells move faster (single cell velocity) if FAs are longer lived?

– Work in HeLa cells has shown that VAP-A depletion leads to higher PI4P levels at the Golgi region (Y. Wakana et al.). Do you think that the observed changes are cell type (CACO2) specific? Have the authors considered the possibility to use other means to measure PI4P levels at the Golgi (e.g. antibody staining)?

– GFP-MAPPER: As I understand it from the reference, there are versions of the GFP-MAPPER plasmid with different lengths of the cytosolic linker part. Could the authors specify which one they used?

– The description of the nocodazole washout experiment is a bit difficult to follow and understand properly in the methods, I think it will help the readers if the authors could explain it a bit clearer.

*Reviewer #2 (Recommendations for the authors):*

The conclusions raised by the authors are convincing. I have listed here some comments, which I hope will strengthen some results and help to improve the manuscript.

Figure 1C: As VAPA KO cells filled the open space slower than control cells and that cell density is obviously lower in VAPA KO cell monolayer, it might be necessary to measure (or discuss) the proliferation rate of VAPA KO cells compared to control cells.

Is there any effect of VAPA depletion on cell-cell contacts required for collective migration?

Figure 2A: the authors focus on the larger FA observed in VAPA KO cells compared to the control cells in the "interior" of the cell. However, from the pictures displayed in Figure 2A, it seems that there is more paxillin signal in the cell periphery of control and VAPA KO + mCh-VAPA KDMD. Is there any effect of VAPA depletion on adhesive structures at the cell contour?

The same comment can be raised for results shown in Figure 5F. Did the authors measure FA dynamics both at the cell center and at the cell periphery? Do the different FA populations have the same behavior in VAPA depletion conditions?

Figure 2C: remove the "e" to "cortactin".

Using data presented in Figure6, the authors claim (lines 201-203) " Thus, we show that VAPA is required for the anchoring of ER-PM contact sites to FA, which happens at the time of FA disassembly"

To consolidate their interpretation, performing the same experiments as the ones conducted for Fig6 C-E but in VAPA KO cells and in VAPA KO cells expressing VAPA construct (rescue experiment) would be necessary.

In the discussion (lines 232-237) and in Figure 7, the authors discuss the role of VAPA-mediated ER-PM contact sites and FA in integrin trafficking. However, in this manuscript, no experiments regarding integrin trafficking were performed. To my opinion, other possible mechanisms might be added to Figure 7 such as the role of VAPA for calcium influx, STIM1 and ORP3.

*Reviewer #3 (Recommendations for the authors):*

Main recommendations

1) Results and hypotheses arising from VAPA KO should be tested by rescue experiments with WT and mutants (such as in Figure 2A). VAPA should be shown at ER-PM contacts in Figure 4 and Fig6 and in the recommended rescue experiments.

2) The data in the paper are from a single Crispr/Cas9 clone of VAPA KO. It is important to confirm some findings with other clones.

3) The direct link between VAPA, PI(4,5P)2 and FA disassembly is missing. The Nir2 (/ORP/lipid transfer pathway) hypothesis should be tested, by KO or siRNA, in cell motility or FA turnover experiment. Integrin endocytosis should be tested in VAPA KO and in rescue experiments. VAPA should be shown at ER-PM contacts near FA with Nir2 or another protein acting on PI(4,5)P2.

4) The authors assessed PI(4)P levels at Golgi and endosomes but not at the PM. As for PI(4,5)P2, it was tested at dorsal PM, but not at the ventral area where FA disassembly occurs. These measurements should be done. For PI(4)P, it would be wise to use the P4M-SiDM probe (PMID 24711504), rather than OSBP-PH which seems highly recruited to the Golgi but not to other membranes. Moreover, previous studies showed that not only VAPB but also VAPA is present at ER-Golgi and ER-endosome contacts in different cell types (by binding to CERT, OSBP, ORPs), with striking effects on PI(4)P modulation. How do the authors explain this apparent discrepancy?

5) For statistics it is highly recommended to use superplots (PMID 32346721). In addition, showing data with only ~10 cells analyzed per condition, as in Figure 3, is hardly acceptable.

Other points:

6) The two sets of images in Fig5E are difficult to compare because the starting point of FA formation is different. The rate of FA assembly appears to be 10 min longer in VAPA KO than in the control, which is not consistent with quantification. It is also difficult to judge the delay in disassembly. The time-lapse images should be comparable by showing the same acquisition times.

7) The authors claim that "contacts" are created between ER-PM contact sites and FA based on a one-pixel overlap (Fig6A, B), but whether the resolution of the image used here allows going that far is questionable.

8) The authors must show the effect on microtubule network during the washout experiment in Fig5F. The loss of FA size is difficult to see based on the images. The bar chart should be replaced by a panel in which individual measurements are shown (see point 5).

– Line 55. VAP not only binds to lipid transfer proteins.

– Line 185. To a non-specialist, it can be difficult to understand why the microtubule network comes into play here.

---

## [Author Response]

Essential Revisions:We all agree that this manuscript reports potentially very interesting findings, but we also think that additional data will be needed to support the claims. The essential revisions requested are the following (1-3 will require new experiments, 4-6 might not, but additional discussion is requested):1) Show additional rescue experiments (and/or test some of the findings with other KO clones). Especially regarding those experiments presented in Figures 1,5, and 6. We all agreed that this is perhaps the major gap in the paper. See detailed reviews below.

Considering the amount of additional experiments requested, we chose to concentrate on the Rescue cell lines. Indeed, they offered the double advantage of testing whether the phenotypes observed in VAPA KO cells are due to VAPA depletion and to interrogate the role of the MSP domain in the different processes studied. We have added new data that include the 2 VAPA KO cell lines expressing either VAPA WT or the mutant VAPA KDMD. Their phenotypes were analyzed in most of the cellular processes studied in the paper and described in figures 1, 2, 5 and 6. See point-by-point responses.

2) Show VAP-A localization by IF together with the localization of GFP-MAPPER MCS.

We have done the experiments requested by the Reviewers concerning that point. As suggested by Reviewer #3, we have added images of endogenous or exogenous VAPA at ER-PM contact sites in figures 4, figure 4—figure supplement 1 and figure 6.

3) Test how peripheral vs. more centrally organized FAs behave with respect to VAP-A.

We have added several data on peripheral FA and investigated the role of VAPA on this subpopulation in figures 2 and 6 (see responses to Reviewer #2).

4) Improve their assessment/discussion on the PI4P and PIP2 results.

We have added some data on PI4P at the PM using another probe and increased the sample size when necessary in figure 3. We have also improved the discussion on our results in the manuscript (see responses to Reviewer #1 and #3).

5) Improve on the method explanation/statistics description, etc.

We have provided all the additional information that were requested by the reviewers.

6) The model is very speculative with respect to the data presented in the manuscript. This is in principle OK, but then it really needs to be discussed more as a speculative/working hypothesis rather than a summary model. While it would be great to be able to experimentally test some of the model's predictions (e.g., test some of the observed effects by KO Nir2, ORP3, or monitor integrin trafficking), we are not requesting any additional experiments in this regard.

As explained in the responses to reviewers, we have removed the model that was too speculative and didn’t give an exhaustive view of the hypotheses. We developed a discussion on the hypotheses in the subsection Ideas and Speculation.

Reviewer #1 (Recommendations for the authors):As mentioned in the public review, I think that this is a very interesting paper, with potentially major findings that will help us understand better the process of cell migration. However, I think that, although parts of the paper are based on solid evidence, there are some important points of the paper that would require additional, stronger, and more compelling evidence.– The VAP-A KO cell line is produced by single clonal selection. I really appreciate the experiments shown in Figure 2A, where the phenotype in the KO was rescued by the expression of WT and not mutant VAP-A. However, similar rescue experiments are not performed elsewhere in the manuscript. I am of course not saying that the rescues are presented in every single experiment, but in some of the key experiments, additional evidence about the specificity of the VAP-A effect should be provided (e.g. have the authors generated additional clonal KOs and tested the effects in those cell lines?)

We have added new data that include the 2 VAPA KO cell lines expressing either VAPA WT or the mutant VAPA KDMD. Their phenotypes were analyzed in most of the cellular processes studied in the paper and described in figures 1, 2, 5 and 6: collective cell migration (Figure 1D-G), cell spreading (Figure 1K-L), organization of actin cytoskeleton (Figure 2D, 2F-G), protrusion dynamics (Figure 2D-E, 2H), focal adhesions size (Figure 2A-C), lifetime (Figure 5D) and proximity with GFP-MAPPER foci (Figure 6A-B). Unfortunately, for technical reasons, it was impossible to quantify the assembly and disassembly rates of focal adhesions in these 2 cell lines (Figure 5). However, the fact that the size and the lifetime of focal adhesions were rescued in VAPA KO+WT but not in VAPA KO+KDMD (Figure 2A-C and Figure 5D) is a proof that the defects of focal adhesion dynamics observed in VAPA KO cells is due to the absence of VAPA and depends on its MSP domain.

– It was not clear to me, based on the image in Figure 2B, that the cortactin signal in the KOs is different from Control. Could the authors quantify that?

We apologize if we were not clear on this part in the manuscript. We do not claim that there is a difference in the grey levels intensity of the cortactin signal between Control and VAPA KO cells. The difference that we observed resides in the distribution of cortactin along the leading edge, in particular the characteristics of cortactin-rich protrusive subdomains that are occupying a higher proportion of the leading edge and are longer in VAPA KO cells (Figure 2F and 2G). In order to clarify these quantifications, we have now added plot profiles of cortactin signal along the leading edge to highlight these cortactin-rich protrusive subdomains (in pink in the figure).

– Line 109: the title of the section reads as "VAPA regulates FAs and actin cytoskeleton through its MSP domain". However, the authors have not tested the role of the MSP domain in the actin effects. Hence, a rescue experiment with the WT and mutant VAP-A would be necessary here.

We apologize if this title seemed overinterpreted: what we called actin cytoskeleton in the title related to the organization of cortactin for which we had rescued conditions. In order to complete this part, we followed the Reviewer’s comment by adding a description of the organization of actin in VAPA KO+WT and VAPA KO+KDMD cell lines (Figure 2D) and quantified the repartition of cortactin and the dynamics of protrusions (Figure 2E-H). As described in the manuscript, all these phenotypes were rescued by the expression of VAPA WT but not VAPA KDMD, indicating that they depend on the MSP domain of VAPA.

– Figure 6: This is a potentially quite cool finding. However, based only on this I'm not sure that the claims in the paper are sustained by the evidence. For instance, is a similar effect as that presented in panel E seen for those FAs, not in contact with an ER-PM MCS (according to panel B, there are about 25% of those)? Similarly, doing this experiment in the KO cells, where there is a fraction (>30%) of FAs proximal to an MCS, would determine whether MCS are enough for this effect or whether VAP-A is indeed playing a role there.

We thank the Reviewer for giving us the opportunity to improve the experimental evidences on this point. We have now added new data that, we hope, will be more convincing (Figure 6C-H).

Concerning the 25-30% of FA not in contact with GFP-MAPPER foci (quantified in Figure 6B), it is possible that they were in fact in contact with a GFP-MAPPER foci during their lifetime, as this quantification reflects the situation at one time point. They might correspond to the FA that are assembling and not yet in contact, as we observe in panels C and E that the first GFP-MAPPER foci appear on pre-existing FA. So we probably took them into account in our dynamic analysis. However, it might also be that a proportion of FA in Control cells are indeed never in contact with GFP-MAPPER foci (or that we are not able to detect them); then for those FA, we won’t be able to answer the Reviewer’s question on the correlation between FA dynamic and GFP-MAPPER dynamic, as there is no GFP-MAPPER signal in the vicinity of the FA.Concerning the 30% of FA in contact with GFP-MAPPER foci in VAPA KO cells, we have now added some data. Our results show that VAPA is required:

Moreover, using TIRF microscopy, we were able to detect the presence of VAPA in GFP-MAPPER foci proximal to FA (Figure 6G-H).

Altogether, these observations point to a specific role of VAPA in the ER-PM contact sites proximal to FA.

– The model is interesting and I think it presents a lot of possibilities for future work. However, as it is now, I think it is very hypothetical (which the authors already state in the legends), and mostly based on published work. This is fine, however, I feel that, given some of the cell-specific differences seen in this report (e.g. Golgi PI4P changes), the ideas in the cartoon might not be 100% extrapolated to their system. Of course, it would be great if the authors test the effects of depleting other ER-PM MCS proteins (Nir2, ORP3), or study integrin uptake, etc. However, I think this would be too much to ask for this paper (so it is not necessary in my opinion), but then this section might need to be integrated as an "Ideas and Speculation" section, according to eLife editorial: https://elifesciences.org/inside-eLife/e3e52a93/eLife-latest-including-ideas-and-speculation-in-eLife-papers

We thank the Reviewer for this suggestion. The part of the Discussion describing potential hypothesis has now been integrated in the “Ideas and Speculation” subsection (Lines 294-338). In addition, the model has been removed from the paper as we felt that it was too speculative and that it doesn’t give an exhaustive view of the possible hypothesis.

Other important points:– Regarding the results in Figure 1C, I, J: The KOs look overall to be larger cells. Is this true? Is this due to clonal selection or because of the KO of VAP-A? Would the authors rescue the phenotype with the re-expression of the WT VAP-A? I think it would be informative to measure the final area/single cell after 24 h of spreading (steady state) and/or measure the nuclear area to be sure these effects are not due to size variability but to VAP-A depletion. Also, images in panel (I) are very pixelated in my PDF.

We have now added new data showing that the expression of VAPA WT in VAPA KO cells partially restores the spreading behaviour of these cells within the first hour, and that it depends on the MSP domain of VAPA (Figure 1K-L), which is in favour of a VAPA-dependent effect rather than a size variability effect. In addition, as requested by the Reviewer, we analyzed the cell areas after 24 hours in the 4 cell lines. As shown in Author response image 1, after 24 hours, the phenotype was inverted compared to 1 hour spreading: the depletion of VAPA induced a strong decrease of the cell area, restored by the expression of exogenous VAPA WT but not VAPA KDMD. This phenotype might be due to the fact that after 24 hours, the cells are not spreading anymore, but have started to move in 2D as individual cells, so their shape and area are intimately linked to their motile behaviour. The small size of VAPA KO cells is correlated to their faster way to move, while Control cells are more static cells with a “fried egg” shape and a high surface spreading. This observation is also well visible in Figure 1H. However, in motile monolayers, VAPA KO cells look bigger than Control cells, which is the opposite of individual motile cells. We think this decrepancy might be due to the fact that in monolayers, the cell area is also influenced by the presence and the integrity of cell-cell junctions that could be altered in VAPA KO cells.

Altogether, our new data clearly show that the size differences are indeed due to the absence of VAPA rather than a size variability due to clonal selection.

**Author response image 1. sa2fig1:** Analysis of cell area 24 hours after plating. A. Epifluorescence images of Control, VAPA KO, VAPA KO+WT and VAPA KO+KDMD cells 24 hours after plating on fibronectin-coated glass and stained as indicated. Scale bar: 100 μm. B. Analysis of cell area 24 hours after plating on fibronectin-coated glass (n=49 to 57 cells from 3 independent experiments). Data were analysed using a One-way Anova Kruskal-Wallis test. (ns: non significant, ***P-values <0.001, **P-values <0.01, *P-values <0.05).

– The values plotted in Figure 3C are surprisingly small if they are indeed reporting at the fluorescence intensity density in the Golgi region relative to the same quantity measured elsewhere in the cytosol. Could the authors double-check if the quantification is done correctly, or maybe I am missing something?

We thank the Reviewer for pointing this out. There was indeed an error in the normalization of the results. We have corrected this error and increased the sample size to more than 20 cells, as suggested by Reviewer #3 (Figure 3C). We obtained a Golgi/Cytosol ratio of around 5 which is in agreement with the PI4P enrichment detected in the Golgi.

– Figure 4E: I think the results are clear, but their representation of them could be improved to help the readers. I'd suggest that the tracking performed in the panels below the TIRF frames are showily more data points, and linked through connecting trajectory lines. Also, explain in the methods how this tracking and the lifetime and speed (panels F, G) have been measured.

As suggested by the Reviewer, we have now improved the representation of the results, by adding more time points and by labeling each time point with a specific colour in the tracking image (Figure 4E). We have also added a small paragraph in the methods to explain the way this analysis was done (Lines 530-534).

– This might be something obvious for the authors/other experts in the field, but why is it that VAP-AKO cells move faster (single cell velocity) if FAs are longer lived?

We thank the Reviewer for raising this point which is not trivial. At that stage, we can’t give any experimental explanation for this, but we can speculate on the reasons why VAPA KO cells have longer-lived FAs and move faster.

First, the group of Denis Wirtz showed, some time ago, that for slow-moving cells (like individual Caco2 cells), an increase in FAs size correlates with a faster cell speed (Kim and Wirtz, 2013), highlighting the fact that having bigger FAs does not necessarily prevent the cells from moving faster.

Second, even if VAPA KO focal adhesions live longer, we have no information on their molecular composition, their maturation/activation status, their connection with the actin-cytoskeleton and their mechano-sensitivity. All these parameters can influence the motile behaviour of the cells, including cell directionality and cell speed (Doyle et al., 2022; Gupton and Waterman-Storer, 2006).

– Work in HeLa cells has shown that VAP-A depletion leads to higher PI4P levels at the Golgi region (Y. Wakana et al.). Do you think that the observed changes are cell type (CACO2) specific?

We thank the Reviewer for mentioning this work by Wakana et al. that we have now added to the manuscript (Lines 77-78). In this study, as in the 3 other studies that we cite in the manuscript (Lines 76-80), the authors depleted both VAPA and VAPB and observed, depending on the cell type and organism, either a reduction (Peretti et al., 2008) or an accumulation (Mao et al., 2019; Wakana et al., 2021) of PI4P levels in the Golgi membranes and its redistribution on endosomes (Dong et al., 2016). The main difference with our study is that we depleted only VAPA. As discussed in lines 159-169 and 265-277, our results showing no effect of VAPA depletion on PI4P at the Golgi or endosomes can be explained by a possible redundancy of VAPA and VAPB in these 2 compartments. In other words, in our VAPA KO cells, the presence of VAPB would be enough to maintain the levels of PI4P at the Golgi and endosomes. This is corroborated by the results of P. di Camilli’s group (Dong et al., 2016) who did not observe any defect in the intra-cellular distribution of PI4P in cells expressing only VAPB (see Figure S1D in their paper).

Moreover, our results show that something different happens at the PM where the presence of VAPB does not seem to be enough to maintain the levels of PI4P and PI(4,5)P2. Interestingly, the same phenotype has been observed in Nir2 depleted cells (discussed in lines 308-311).

Have the authors considered the possibility to use other means to measure PI4P levels at the Golgi (e.g. antibody staining)?

We tried to stain our cells with anti-PI4P antibodies but didn’t get a sufficient signal to be able to quantify properly the images. However, as requested by Reviewer #3, we have now added new data on the amount of PI4P at the PM using P4M-SidM probe and showed that PI4P levels at the PM are decreased in VAPA KO cells, similar to PI(4,5)P2 levels (Figure 3D).

GFP-MAPPER: As I understand it from the reference, there are versions of the GFP-MAPPER plasmid with different lengths of the cytosolic linker part. Could the authors specify which one they used?

We used the long version of MAPPER that has been characterized throughout Chang et al. paper (Chang et al., 2013), and that contains two flexible helical linkers, (EAAAR) 4 and (EAAAR) 6, upstream and downstream flanking regions of the FRB, respectively. This precision has been added to the Methods section (Lines 364-367).

The description of the nocodazole washout experiment is a bit difficult to follow and understand properly in the methods, I think it will help the readers if the authors could explain it a bit clearer.

We have now explained more precisely the nocodazole wash-out experiment in the Methods (Lines 467-476).

Reviewer #2 (Recommendations for the authors):The conclusions raised by the authors are convincing. I have listed here some comments, which I hope will strengthen some results and help to improve the manuscript.Figure 1C: As VAPA KO cells filled the open space slower than control cells and that cell density is obviously lower in VAPA KO cell monolayer, it might be necessary to measure (or discuss) the proliferation rate of VAPA KO cells compared to control cells.

We thank the Reviewer for giving us the opportunity to clarify this point. Concerning the proliferation rate, in order to avoid any proliferation bias, we inhibited cell division by treating the cells with mitomycin. This information has been added in the Results section (Lines 104-105). When VAPA KO cells are displacing collectively, they spread more than Control cells (see point 6 of Reviewer #1) which is why, on the figure 1D, the density looks lower.

Is there any effect of VAPA depletion on cell-cell contacts required for collective migration?

We have not investigated the organization of cell-cell junctions in VAPA KO cells. It is indeed highly probable that VAPA regulates junctional organization, as PI(4,5)P2 has been shown to control endocytosis of cadherins and organization of the actin cortex. So the phenotype observed in collective migration might be due, at least in part, to a function of VAPA at cell-cell contacts. However, as we observe the same motility phenotype on individual cells (Figure 1H-J), we can be confident on the fact that the role of VAPA on cell motility is at least partially independent of cell-cell junctions, as mentioned in Lines 109-112.

Figure 2A: the authors focus on the larger FA observed in VAPA KO cells compared to the control cells in the "interior" of the cell. However, from the pictures displayed in Figure 2A, it seems that there is more paxillin signal in the cell periphery of control and VAPA KO + mCh-VAPA KDMD. Is there any effect of VAPA depletion on adhesive structures at the cell contour?The same comment can be raised for results shown in Figure 5F. Did the authors measure FA dynamics both at the cell center and at the cell periphery? Do the different FA populations have the same behavior in VAPA depletion conditions?

We thank the Reviewer for raising this point. We have now added some data on peripheral FA that, in fact, correspond to the population of nascent adhesions and focal complexes. As shown in Fig. 2A-C and Fig. 6A-B, we observed no difference between the cell lines, or a very slight one, in the size of peripheral FA and their proximity with GFP-MAPPER foci. These results suggest that VAPA is not involved in the first steps of FA assembly, which is consistent with the observation that Control and VAPA KO FA assemble at the same speed (Fig. 5B), and that GFP-MAPPER foci appear in the vicinity of FA after their assembly (Fig. 6C-D).

Figure 2C: remove the "e" to "cortactin".

This error has been corrected.

Using data presented in Figure6, the authors claim (lines 201-203) " Thus, we show that VAPA is required for the anchoring of ER-PM contact sites to FA, which happens at the time of FA disassembly"To consolidate their interpretation, performing the same experiments as the ones conducted for Fig6 C-E but in VAPA KO cells and in VAPA KO cells expressing VAPA construct (rescue experiment) would be necessary.

We have now added new data in Figure 6, showing that in VAPA KO cells, there is no preferential arrival of the 1^st^ GFP-MAPPER foci before FA disassembly (Figure 6C-E), and that the proximity between FA and GFP-MAPPER foci lasts shorter (Figure 6F). Unfortunately, because of technical issues with our new TIRF microscope, we were not able to include Rescue cell lines in these experiments that necessitated more acquisitions. However, we added some data on the Rescue cell lines in Figure 6A-B showing that VAPA is indeed necessary for the proximity between central FA and GFP-MAPPER.

In the discussion (lines 232-237) and in Figure 7, the authors discuss the role of VAPA-mediated ER-PM contact sites and FA in integrin trafficking. However, in this manuscript, no experiments regarding integrin trafficking were performed. To my opinion, other possible mechanisms might be added to Figure 7 such as the role of VAPA for calcium influx, STIM1 and ORP3.

We fully agree with the Reviewer, which is why we removed the model that we felt was too speculative and didn’t give an exhaustive view of the possible hypothesis. We also integrated the part of the Discussion describing potential hypothesis in the “Ideas and Speculation” subsection (Lines 294338). The possible connection with STIM1 and ORP3 is discussed in this subsection.

Reviewer #3 (Recommendations for the authors):Main recommendations1) Results and hypotheses arising from VAPA KO should be tested by rescue experiments with WT and mutants (such as in Figure 2A). VAPA should be shown at ER-PM contacts in Figure 4 and Fig6 and in the recommended rescue experiments.

We thank the Reviewer for raising these points and giving us the opportunity to improve the paper. As explained in our responses to Reviewer #1, we have now added a significant amount of data in the 2 VAPA KO cell lines expressing either VAPA WT or the mutant VAPA KDMD. Their phenotypes were analyzed in most of the cellular processes studied in the paper and described in figures 1, 2, 5 and 6: collective cell migration (Figure 1D-G), cell spreading (Figure 1K-L), organization of actin cytoskeleton (Figure 2D, 2F-G), protrusion dynamics (Figure 2D-E, 2H), focal adhesions size (Figure 2A-C), lifetime (Figure 5D) and proximity with GFP-MAPPER foci (Figure 6A-B).

VAPA should be shown at ER-PM contacts in Figure 4 and Fig6 and in the recommended rescue experiments.

As suggested by the Reviewer, we have added images of endogenous or exogenous VAPA at ER-PM contact sites:

– by confocal microscopy in Control cells transiently expressing Cherry-VAPAWT and GFP-MAPPER (Figure 4C)

– by confocal microscopy in Control cells transiently expressing GFP-MAPPER and immuno-stained with anti-VAPA antibodies (Figure 4—figure supplement 1A)

In addition, in Figure 6G, we added a TIRF image showing the localization of exogenous VAPAWT expressed in VAPA KO cells co-localizing with ER-PM contact sites close to focal adhesions, which reinforces our results showing the local function of VAPA at focal adhesions.

2) The data in the paper are from a single Crispr/Cas9 clone of VAPA KO. It is important to confirm some findings with other clones.

The editors requested additional data in Rescue cell lines and/or other VAPA KO clones. Considering the difficulties we’ve had during this review process (in particular the departure of the first author and the technical issues with our TIRF microscope) and the amount of additional experiments requested, we chose to concentrate on the Rescue cell lines. Indeed, they offered the double advantage of testing whether the phenotypes observed in VAPA KO cells are due to VAPA depletion and to interrogate the role of the MSP domain in the different processes studied.

3) The direct link between VAPA, PI(4,5P)2 and FA disassembly is missing. The Nir2 (/ORP/lipid transfer pathway) hypothesis should be tested, by KO or siRNA, in cell motility or FA turnover experiment. Integrin endocytosis should be tested in VAPA KO and in rescue experiments. VAPA should be shown at ER-PM contacts near FA with Nir2 or another protein acting on PI(4,5)P2.

We agree with the Reviewer that our paper doesn’t give any experimental evidence on the molecular mechanisms involved in the function of VAPA during cell motility. However, we feel that these elements would be beyond the scope of this work, as mentioned by the Editors.

4) The authors assessed PI(4)P levels at Golgi and endosomes but not at the PM. As for PI(4,5)P2, it was tested at dorsal PM, but not at the ventral area where FA disassembly occurs. These measurements should be done. For PI(4)P, it would be wise to use the P4M-SiDM probe (PMID 24711504), rather than OSBP-PH which seems highly recruited to the Golgi but not to other membranes.

As suggested by the Reviewer, we have now added some data on PI(4)P using the P4M-SiDM probe (Figure 3D-E). As for PI(4, 5)P2, we observed lower amounts of PI(4)P at the plasma membrane of VAPA KO cells. We have tried to assess the local regulation of PI(4,5)P2 and PI(4)P on the ventral area where FA disassembly occurs, but we were not able to see any variations, probably because the events we were trying to catch are very transient.

Moreover, previous studies showed that not only VAPB but also VAPA is present at ER-Golgi and ER-endosome contacts in different cell types (by binding to CERT, OSBP, ORPs), with striking effects on PI(4)P modulation. How do the authors explain this apparent discrepancy?

Our finding that VAPA depletion doesn’t have any effect on PI(4)P at the Golgi is, in fact, not in contradiction with published work. Indeed, in the 4 studies that we cite in the manuscript (Lines 7680), the authors depleted both VAPA and VAPB and observed, depending on the cell type and organism, either a reduction (Peretti et al., 2008) or an accumulation (Mao et al., 2019; Wakana et al., 2021) of PI4P levels in the Golgi membranes and its redistribution on endosomes (Dong et al., 2016). The main difference with our study is that we depleted only VAPA. As discussed in lines 159-169 and 265-277, our results can be explained by a possible redundancy of VAPA and VAPB in these 2 compartments. In other words, in our VAPA KO cells, the presence of VAPB would be enough to maintain the levels of PI4P at the Golgi and endosomes. This is corroborated by the results of P. di Camilli’s group (Dong et al., 2016) who did not observe any defect in the intra-cellular distribution of PI4P in cells expressing only VAPB (see Figure S1D in their paper).

5) For statistics it is highly recommended to use superplots (PMID 32346721). In addition, showing data with only ~10 cells analyzed per condition, as in Figure 3, is hardly acceptable.

Most of the data that were represented as bar graphs have been modified to Scatter plots or Box and Whiskers, to allow the visualization of individual events. As suggested by the Reviewer, we increased the number of cells analyzed in Figure 3C.

Other points:6) The two sets of images in Fig5E are difficult to compare because the starting point of FA formation is different. The rate of FA assembly appears to be 10 min longer in VAPA KO than in the control, which is not consistent with quantification. It is also difficult to judge the delay in disassembly. The time-lapse images should be comparable by showing the same acquisition times.

We thank the Reviewer for pointing out this mistake. As suggested, we have now represented image sequences with the same starting point of FA formation and the same acquisition times in Control and VAPA KO cells (Figure 5A). They are also more representative of the phenotypes observed in the quantification.

7) The authors claim that "contacts" are created between ER-PM contact sites and FA based on a one-pixel overlap (Fig6A, B), but whether the resolution of the image used here allows going that far is questionable.

We fully agree with the Reviewer that “contacts” is not the appropriate word here, considering the resolution used. We have now removed this word from the manuscript. Using a super resolutive SIM microscope, we were able to determine that GFP-MAPPER foci and FA were distant of around 50150nm (Figure 6—figure supplement 1A). Moreover, the analysis of GFP-MAPPER foci dynamics revealed that in Control cells, they persisted several minutes (in average 6 minutes) in the vicinity of FA (Figure 6F), suggesting that they were either directly or indirectly anchored to FA.

8) The authors must show the effect on microtubule network during the washout experiment in Fig5F. The loss of FA size is difficult to see based on the images. The bar chart should be replaced by a panel in which individual measurements are shown (see point 5).

We have now added images of the microtubules network at different time points of the washout experiment (Figure 5—figure supplement 1A). They show a relatively fast recovery of the microtubule network after washout of nocodazole. As suggested by the Reviewer, we put more representative images of FA staining in Figure 5E and replaced the bar graph by Scatter plots in Figure 5F.

– Line 55. VAP not only binds to lipid transfer proteins.

We have modified the 2 sentences in question (Lines 57-62).

– Line 185. To a non-specialist, it can be difficult to understand why the microtubule network comes into play here.

We have now added a sentence explaining why we addressed the role of microtubules (Lines 216221)

References

Chang CL, Hsieh TS, Yang TT, Rothberg KG, Azizoglu DB, Volk E, Liao JC, Liou J. 2013. Feedback regulation of receptor-induced Ca^2+^ signaling mediated by e-syt1 and nir2 at endoplasmic reticulum-plasma membrane junctions. *Cell Rep* 5:813–825. doi:10.1016/j.celrep.2013.09.038

Dong R, Saheki Y, Swarup S, Lucast L, Harper JW, De Camilli P. 2016. Endosome-ER Contacts Control Actin Nucleation and Retromer Function through VAP-Dependent Regulation of PI4P. Cell 166:408–423. doi:10.1016/j.cell.2016.06.037

Doyle AD, Nazari SS, Yamada KM. 2022. Cell-extracellular matrix dynamics. Phys Biol 19. doi:10.1088/1478-3975/AC4390

Gupton SL, Waterman-Storer CM. 2006. Spatiotemporal Feedback between Actomyosin and FocalAdhesion Systems Optimizes Rapid Cell Migration. *Cell* 125:1361–1374. doi:10.1016/j.cell.2006.05.029

Kim DH, Wirtz D. 2013. Focal adhesion size uniquely predicts cell migration. *FASEB J* 27:1351–1361. doi:10.1096/FJ.12-220160

Mao D, Lin G, Tepe B, Zuo Z, Tan KL, Senturk M, Zhang S, Arenkiel BR, Sardiello M, Bellen HJ. 2019. VAMP associated proteins are required for autophagic and lysosomal degradation by promoting a PtdIns4P-mediated endosomal pathway. *Autophagy* 15:1214–1233. doi:10.1080/15548627.2019.1580103

Peretti D, Dahan N, Shimoni E, Hirschberg K, Lev S. 2008. Coordinated lipid transfer between the endoplasmic reticulum and the golgi complex requires the VAP proteins and is essential for Golgi-mediated transport. *Mol Biol Cell* 19:3871–3884. doi:10.1091/mbc.E08-05-0498

Wakana Y, Hayashi K, Nemoto T, Watanabe C, Taoka M, Angulo-Capel J, Garcia-Parajo MF, Kumata H, Umemura T, Inoue H, Arasaki K, Campelo F, Tagaya M. 2021. The er cholesterol sensor scap promotes carts biogenesis at er–golgi membrane contact sites. *J Cell Biol* 220. doi:10.1083/JCB.202002150/VIDEO-1